

# Restoration of the non-Hermitian bulk-boundary correspondence via topological amplification

**Matteo Brunelli[1], Clara C. Wanjura[2,3] and Andreas Nunnenkamp[4]**

1 Department of Physics, University of Basel, Klingelbergstrasse 82, 4056 Basel, Switzerland
2 Cavendish Laboratory, University of Cambridge, Cambridge CB3 0HE, United Kingdom
3 Max Planck Institute for the Science of Light, Staudtstraße 2, 91058 Erlangen, Germany
4 Faculty of Physics, University of Vienna, Boltzmanngasse 5, 1090 Vienna, Austria

## Abstract

Non-Hermitian (NH) lattice Hamiltonians display a unique kind of energy gap and extreme sensitivity to boundary conditions. Due to the NH skin effect, the separation between edge and bulk states is blurred and the (conventional) bulk-boundary correspondence is lost. Here, we restore the bulk-boundary correspondence for the most paradigmatic class of NH Hamiltonians, namely those with one complex band and without symmetries. We obtain the desired NH Hamiltonian from the mean-field evolution of driven-dissipative cavity arrays, in which NH terms—in the form of non-reciprocal hopping amplitudes, gain and loss—are explicitly modeled via coupling to (engineered and non-engineered) reservoirs. This approach removes the arbitrariness in the definition of the topological invariant, as point-gapped spectra differing by a complex-energy shift are not treated as equivalent; the origin of the complex plane provides a common reference (base point) for the evaluation of the topological invariant. This implies that topologically non-trivial Hamiltonians are only a strict subset of those with a point gap and that the NH skin effect does not have a topological origin. We analyze the NH Hamiltonians so obtained via the singular value decomposition, which allows to express the NH bulk-boundary correspondence in the following simple form: an integer value $\nu$ of the topological invariant defined in the bulk corresponds to $|\nu|$ singular vectors exponentially localized at the system edge under open boundary conditions, in which the sign of $\nu$ determines which edge. Non-trivial topology manifests as directional amplification of a coherent input with gain exponential in system size. Our work solves an outstanding problem in the theory of NH topological phases and opens up new avenues in topological photonics.

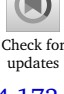
doi:10.21468/SciPostPhys.15.4.173

# 1 Introduction

A universal feature of topological phases is the presence of states localized at the boundaries, as a result of the non-trivial topology of the bulk. This argument is formalized in the celebrated bulk-boundary correspondence (BBC), which expresses a one-to-one correspondence between the values of a topological invariant constructed from the Bloch states of an infinite periodic system and the number of edge modes in a finite system [1,2]. The BBC provides the foundations of our understanding of topological states of matter in systems described by Hermitian Hamiltonians [3]. Recently, a new and exciting line of inquiry has emerged, which is concerned with extending these considerations to systems described by non-Hermitian (NH) Hamiltonians [4–6]. NH Hamiltonians are a powerful tool to model the evolution of open systems in contact with an environment [7].

Lattice models described by NH Hamiltonians display novel and often exotic phenomena with no Hermitian counterpart. Among those is a unique kind of energy gap, known as point gap [5,8], which occurs when the spectrum winds in the complex energy plane as

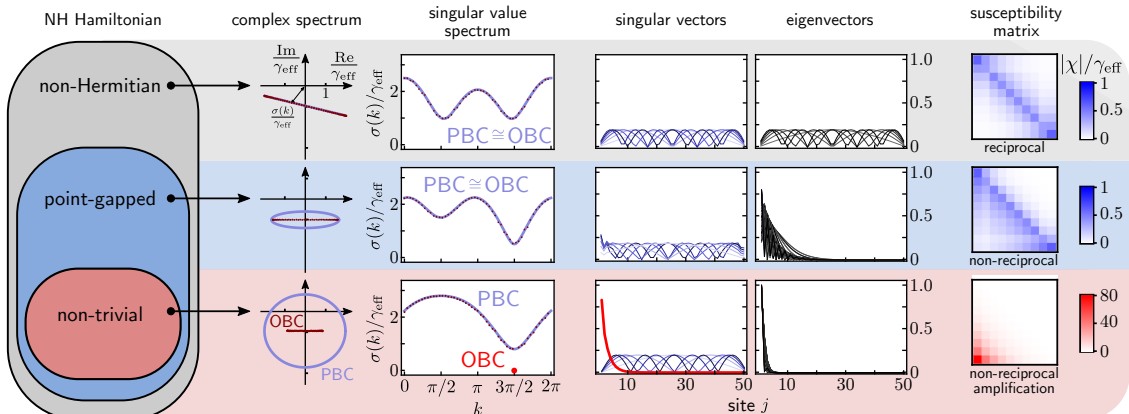

Figure 1: **Overview of the non-Hermitian bulk-boundary correspondence.** The non-Hermitian (NH) Hamiltonian is a realization of the Hatano-Nelson model in a driven-dissipative cavity array [see Eqs. (10) and (11) for $L = 1$]. From the complex spectrum of the associated Bloch Hamiltonian Eq. (1), we can distinguish three distinct regimes, i.e., those with no point gap (degenerate spectrum), point-gapped trivial, and point-gapped non-trivial. These form nested sets, as shown in the Venn diagram on the left. For each of these three regimes we compare several quantities, arranged in different columns. From left to right: the complex spectrum under periodic boundary conditions (PBC, solid line) and open boundary conditions (OBC, dots); the spectrum of singular values under PBC (solid line) and OBC (dots); some representatives of the left singular vectors (in absolute value) including the localized zero singular vector in the topologically non-trivial regime (red thick curve); some representatives of the left eigenvectors (in absolute value); the susceptibility matrices characterizing the response of each site to a weak coherent probe, which describe the photon transmission under OBC (the magnitude of the response is indicated by the color bar on the right; note the presence of amplification in the topologically non-trivial case). The expressions of the coefficients in Eq. (1) are given by $\mu_0/\gamma_{\text{eff}} = -i$, $\mu_{\pm 1}/\gamma_{\text{eff}} = \frac{1}{2}\left(\Lambda - i\mathcal{C}e^{\mp i\theta}\right)$ [see Eqs. (15) and (16)] with the following values of the parameters. Top row: $(\theta, \Lambda, \mathcal{C}) = (0, 2, 0.5)$; middle row: $(\theta, \Lambda, \mathcal{C}) = (\pi/2, 2, 0.5)$; bottom row: $(\theta, \Lambda, \mathcal{C}) = (\pi/2, 2, 1.8)$. In all panels, the OBC quantities are computed for a finite-size system with $N = 50$.

the quasi-momentum is scanned across the Brillouin zone (BZ), and an extreme sensitivity to changes of boundary conditions [9–12]. In fact, NH Hamiltonians can display a striking discrepancy in their spectrum under periodic boundary conditions (PBC) and open boundary conditions (OBC), accompanied by an extensive number of eigenvectors that localize at the system edges under OBC, a phenomenon known as non-Hermitian skin effect (NHSE) [9, 13]. For instance, in the celebrated Hatano-Nelson model (without disorder) [14, 15], all right eigenvectors are exponentially localized at one edge of the system and all left eigenvectors at the other edge. More generally, it has been established that one-dimensional NH Hamiltonians featuring a point gap in their spectrum—or point-gapped Hamiltonians for short—show the NHSE [16,17]. The fact that an extensive number of bulk modes localize at the system boundary undermines the BBC, leading to what is known as the breakdown of the (conventional) BBC in NH systems.

Major efforts have been made to *modify* the BBC in order to accommodate the unconventional features brought about by the NHSE. Notable attempts to restore the BBC include the introduction of a generalized Brillouin zone and non-Bloch band theory [13, 18–21] and an approach based on bi-orthogonal quantum mechanics [10, 22, 23]. These approaches provide

deep insights into several aspects of the topology of NH Hamiltonians but at the same time require revisiting the cornerstones of modern solid-state physics, such as the Bloch theorem. Moreover, the nature of the BBC for point-gapped Hamiltonians, which cannot be reduced to (or deduced from) a limiting case in which the conventional BBC holds [4], still remains an open question.

In this work we reinstate the BBC for the most paradigmatic class of NH lattice models, namely those featuring a single band with a point gap and no symmetry. We show that, akin to Hermitian systems, a one-to-one correspondence between the bulk and the boundary holds also for NH topological systems, which is expressed as follows:

*An integer value $\nu \in \mathbb{Z}$ of the winding number defined on the complex spectrum of the system under periodic boundary conditions corresponds to $|\nu|$ exponentially small singular values associated with singular vectors that are exponentially localized at the system edge under open boundary conditions and vice versa; the sign of $\nu$ determines at which edge the vectors localize.*

Each of these singular vectors is endowed with the following properties: (i) it corresponds to a vanishing (exponentially in system size) singular value, i.e., it is a zero mode; we hence refer to it as a *zero singular mode*. (ii) It is exponentially localized at the boundary (with left and right singular vectors being localized at opposite ends), i.e., it is an edge mode. (iii) The pair of left and right singular vectors possess a well-defined chirality, dictated by the sign of $\nu$. We then see that the zero singular modes possess all the defining properties of edge states. Unlike eigenvectors, however, they do not experience the NHSE and are counted correctly by the winding number, i.e., their number coincides with the value predicted by the bulk topological invariant. This is the essence of the BBC for NH systems.

Our formulation of the BBC relies on two key ingredients. The first one is the singular value decomposition (SVD), which is instrumental to recover the correspondence. Whenever dealing with point-gapped Hamiltonians, we will show that it is the SVD, rather than the standard eigendecomposition, to faithfully describe their properties. Ref. [24] employed the SVD to determine the topological properties of the Hatano-Nelson model via a mapping to the Hermitian Su-Schrieffer-Heeger (SSH) model, provided analytical expressions for singular values and vectors, and studied dynamical stability analytically and robustness to disorder numerically. However, no connection to BBC was drawn. The SVD was applied to the topology of a NH Su-Schrieffer-Heeger (SSH) model and a NH Chern insulator in Ref. [25], where a connection to the BBC was made. Since the NH Hamiltonians we are interested in here are non-normal, for which the eigendecomposition may become inadequate, pseudospectra have also been considered alongside the SVD [5, 26, 27].

The second ingredient concerns the specification of the Hamiltonians entering the correspondence. Indeed, the NH Hamiltonians that reveal the BBC are those encoded in the mean-field evolution of driven-dissipative cavity arrays [24, 28, 29]. We start from the description of the underlying open quantum system (in order to model explicitly both engineered and non-engineered dissipative processes) and study the dynamics of the classical amplitudes. In particular, non-reciprocal photon hopping is implemented by means of a reservoir engineering approach [30, 31]. Moreover, in our approach, NH topology is revealed by the system's response to an external probe, which naturally introduces a frequency reference. This directly impacts the topological properties of the system. In fact, we find that the complex spectra are not invariant under complex-energy shifts: the origin, which separates decaying from amplifying dynamics and detuned from resonant probes, provides a fixed reference (base point) for the evaluation of the topological invariant.

Although our formulation of the BBC has the same formal structure as that of Hermitian topological insulators, the nature of NH topological phases is completely different. NH topology manifests as $|\nu|$ channels of directional amplification, each characterized by a gain that increases *exponentially* with system size. This behavior is unique to the edge states under

OBC and we refer to it as NH topological amplification. This is not to be confused with lasing, which has also been investigated in several topological systems [32–34], the key difference being that topological amplification relies on linear equations of motion. Remarkably, despite the presence of amplification in non-trivial regimes, moving to OBC can render the system stable. NH topological phases correspond to stable stationary regimes under OBC and NH topological phase transitions are transitions between steady states [24, 28]. Ref. [24] studied topological amplification in photonic lattices with nearest-neighbor coupling and used the SVD to obtain a mapping between the NH Hamiltonian and the eigensystem of a doubled Hermitian Hamiltonian. In this way, they established a connection between amplification in the system response and the standard theory of topological insulators and predicted zero singular modes in topological non-trivial regimes. In Ref. [28] we unveiled a one-to-one correspondence between non-trivial NH topology and directional amplification. Directional amplification in NH lattices has also been studied in the context of non-Bloch band theory [35] and in the topologically non-trivial regime a novel kind of metastability has been predicted [27].

Different from previous studies, here we consider one-dimensional lattices with arbitrary long-range coupling, which is key to formulate the BBC for arbitrary integer values, clarify the connection with point-gap topology, and disentangle the role of the non-Hermitian skin effect (NHSE) both from amplification and nontrivial NH topology.

Our ideas are directly relevant for applications to driven-dissipative lattice systems where aspects of NH physics (topology, nonreciprocal couplings, NHSE) have already been demonstrated, e.g. in photonic systems [36–39], topolectric circuits [40–44], exciton polariton lattices [45, 46], mechanical [47, 48] and robotic [49] metamaterials. Especially suitable for implementing our ideas are nano-optomechanical lattices [50, 51] and superconducting circuit optomechanics [52], where non-reciprocal couplings, gain and loss can be engineered to a high degree. We also expect our approach to be applicable to more general NH systems, e.g. with symmetries, multiple bands or higher dimensions, and to provide an ideal starting point for investigating NH topology in open quantum systems [53–56]. Finally, our framework is directly relevant for sensing applications in NH lattices [57–59] and for designing novel directional amplifiers [60–63].

The rest of this work is structured as follows. In Sec. 2 we give an overview of the key steps using the Hatano-Nelson model as a case study. In Sec. 3 we introduce the class of systems studied in this work, which are driven-dissipative cavity arrays coupled to engineered and non-engineered reservoirs. In Sec. 4 we show how the desired unconditional NH Hamiltonian can be derived from the master equation of the driven-dissipative system. In Sec. 5 we address the implications of our NH Hamiltonian for the classification of NH topological phases. In Sec. 6 we investigate the properties underpinning the opening of a point gap in the complex spectrum, establishing a connection with non-normality and non-reciprocity under OBC. In Sec. 7 we introduce the SVD and show how it leads to a notion of gap closure and reopening, signaling a topological phase transition for point-gapped spectra. In Sec. 8 we show, for the concrete case of the Hatano-Nelson model, how the SVD correctly counts the number of boundary modes under OBC. In Sec. 9 we prove the BBC for NH systems in a general way, by establishing a mapping to a generalized Hermitian SSH model. In Sec. 10 we show how bulk non-trivial topology manifests itself as NH topological amplification under OBC. In Sec. 11 we discuss the robustness against disorder of NH topological phases. In Sec. 12 we show how, within our framework, the NHSE is not tied to a topological origin. Finally, Sec. 13 contains our conclusions and some perspectives for future investigations.

# 2 Overview of the non-hermitian bulk-boundary correspondence: The Hatano-Nelson model

We start by providing an overview of the key steps leading to the BBC, illustrated in Fig. 1 in terms of the simplest non-trivial model, namely an implementation of the Hatano-Nelson model in an array of coupled cavities [24, 28]. As shown in Sec. 3, we engineer the system in such a way that the mean field amplitudes (here in quasi-momentum space) evolve according to the NH Bloch Hamiltonian

$$H(k) = \mu_0 + \mu_{-1} e^{-ik} + \mu_1 e^{ik}. \tag{1}$$

The expressions of the complex coefficients $\mu_0$, $\mu_{\pm 1}$ depend on the details of both the Hamiltonian and the reservoirs, see Eqs. (15) and (16). The presence of the constant term $\mu_0$ is the main difference with the standard Hatano-Nelson model of non-reciprocal hopping $H_{\text{HN}} = H(k) - \mu_0$, commonly employed in the literature [5]. The model in Eq. (1) displays the following relevant features:

**1. Three distinct regimes:** Unlike $H_{\text{HN}}$, it has three distinct regimes, highlighted in the three rows of Fig. 1. For $|\mu_{-1}| = |\mu_1|$ (top row), the hopping is reciprocal and there is no point gap in the spectrum. For $|\mu_{-1}| \neq |\mu_1|$, the hopping becomes non-reciprocal, which determines the opening of a point gap, i.e., $H(k)$ describes a curve in the complex plane with an interior as $k$ is scanned across the BZ. A point-gapped spectrum can be further characterized as topologically trivial ($\nu = 0$, middle row) or non-trivial ($\nu = \pm 1$, bottom row), depending on the value of the winding number $\nu$, see Eq. (19); we see that the latter condition is achieved when $H(k)$ encircles the origin [24, 27, 28].

**2. Inequivalence between point gap and non-trivial topology:** The key difference with respect to $H_{\text{HN}}$ is that, for a point-gapped spectrum, Eq. (1) allows for both topologically trivial and non-trivial states. In our framework, the set of topological Hamiltonians is a strict subset of those with a point gap, as shown in the Venn diagram. This is due to the fact that $\mu_0$ removes the invariance of $H(k)$ under complex shifts. This fixes the value of the topological invariant, which is computed with respect to the origin, rather than to an arbitrary base point.

**3. From complex spectrum to singular value spectrum:** With the origin providing a common reference, the distance of the complex spectrum to the origin $|H(k)| \equiv \sigma(k)$ defines a legitimate bandstructure, in terms of the singular values $\sigma(k)$, which we call *singular value spectrum*, see Eq. (27). For a point-gapped spectrum, a transition from/to a NH topological phase ($\nu = \pm 1$) is accompanied by the closure and reopening of a real-valued gap in $\sigma(k)$ [25], which we call *non-Hermitian gap*, see Eq. (28). Under OBC, a non-trivial phase (here $\nu = -1$) is signaled by the appearance of *a single* zero singular value (exponentially small in system size) in the BZ, while the rest of the spectrum does not deviate from $\sigma(k)$ and remains gapped [24, 25]. Therefore, the number of zero singular values coincides with the absolute value of the winding number and this is the signature of the restored BBC on the level of the singular value spectrum.

**4. The non-Hermitian skin effect is not topological:** By inspecting the singular vectors (we plot the left ones) under OBC, we see that almost all of them remain delocalized across all three regimes, and thus the bulk is left intact. Only in the non-trivial regime (here $\nu = -1$), we find *a single* zero singular mode, corresponding to the zero singular value, which is exponentially localized at the left boundary of the finite-size system, see Eqs. (31) and (32). This is the signature of the restored BBC at the level of the singular vectors. In contrast, the left eigenvectors display NHSE, both for trivial and non-trivial configurations alike.

**5. NH topology corresponds to directional amplification:** Each regime reveals distinct transport properties, shown in the rightmost column in terms of the OBC system's response to a input coherent probe, and quantified by the on-resonance susceptibility matrix

$|\chi(0)| = |H^{-1}|$. From the matrix plots, we see that the regime $|\mu_{-1}| = |\mu_1|$ leads to reciprocal transport, $|\mu_{-1}| \neq |\mu_1|$, $\nu = 0$ to directional (non-reciprocal) transport with near-unit gain, while $|\mu_{-1}| \neq |\mu_1|$, $\nu \neq 0$ leads to directional transport with exponential gain (in system size) [24, 28]. Physically, $\nu \neq 0$ is achieved by introducing gain via the term $\mu_0$. The absence of positive imaginary parts of the spectrum under OBC allows to characterize NH topology as steady-state directional amplification [24, 28].

## 3  Model

The system we consider is a one-dimensional array of $N$ cavity modes $\hat{a}_m$, in which the coupling among different sites is mediated by both Hamiltonian processes and dissipative processes. The evolution of the system is formally described by the Lindblad master equation ($\hbar = 1$)

$$\dot{\varrho} = -\mathrm{i}\big[\hat{\mathcal{H}}, \hat{\varrho}\big] + \sum_{m,n} L_{mn}\big(\hat{a}_m \hat{\varrho} \hat{a}_n^\dagger - \tfrac{1}{2}\{\hat{a}_n^\dagger \hat{a}_m, \hat{\varrho}\}\big) + \sum_{m,n} G_{mn}\big(\hat{a}_m^\dagger \hat{\varrho} \hat{a}_n - \tfrac{1}{2}\{\hat{a}_n \hat{a}_m^\dagger, \hat{\varrho}\}\big), \quad (2)$$

where $\hat{\mathcal{H}}$ is the Hamiltonian of the system and the second (third) term on the right-hand side describes loss (gain) processes due to coupling to reservoirs, with coupling matrix $L$ ($G$); the correlated emission (absorption) of photons from different cavities, with $[\hat{a}_m, \hat{a}_n^\dagger] = \delta_{mn}$, mediates a dissipative coupling among them. We will be interested in the evolution of the mean cavity amplitudes $\langle \hat{a}_m \rangle \equiv \alpha_m$, which is given by

$$\dot{\alpha}_m = \mathrm{i}\big\langle\big[\hat{\mathcal{H}}, \hat{a}_m\big]\big\rangle - \sum_n \left(\frac{L_{mn}^* - G_{mn}}{2}\right)\alpha_n, \quad (3)$$

where the expectation value is taken over the state of the system $\hat{\varrho}$.

The Hamiltonian of the system $\hat{\mathcal{H}} = \hat{\mathcal{H}}_0 + \hat{\mathcal{H}}_J + \hat{\mathcal{H}}_d$ consists of the following terms

$$\hat{\mathcal{H}}_0 = \sum_{m=1}^N \omega_\mathrm{c} \hat{a}_m^\dagger \hat{a}_m, \quad (4)$$

$$\hat{\mathcal{H}}_J = \sum_{\ell=1}^L \sum_{m=1}^{N-\ell} \big(J_\ell \hat{a}_m^\dagger \hat{a}_{m+\ell} + \mathrm{H.c.}\big), \quad (5)$$

$$\hat{\mathcal{H}}_d = -\mathrm{i} \sum_{m=1}^N \big(\Omega_m(t) \hat{a}_m^\dagger - \mathrm{H.c.}\big). \quad (6)$$

The first term describes free oscillations of each cavity with frequency $\omega_\mathrm{c}$, which we assume to be the same for all the cavities. The second describes photon hopping between cavities with range up to $L$ sites and real amplitudes $\{J_\ell\}_{\ell=1}^L$. The third describes probing of the cavities by a weak drive, each cavity being coupled to an input-output waveguide at a rate $\gamma$ and probed via the input field $\langle \hat{a}_{\mathrm{in},m}(t)\rangle$, and hence displaced by $\Omega_m(t) = \sqrt{\gamma}\langle \hat{a}_{\mathrm{in},m}(t)\rangle$ [64]. The coupling matrices characterizing the dissipative part are given by

$$L_{mn} = \left(\gamma + 2\sum_{\ell=1}^L \Gamma_\ell\right)\delta_{mn} + \sum_{\ell=1}^L \Gamma_\ell \big(e^{\mathrm{i}\theta_\ell}\delta_{m,n-\ell} + e^{-\mathrm{i}\theta_\ell}\delta_{m-\ell,n}\big), \quad (7)$$

$$G_{mn} = \kappa\delta_{mn}. \quad (8)$$

In Eq. (7) we can distinguish between two contributions: the non-engineered photon decay to input-output waveguide at each site, occurring at a rate $\gamma$, and the decay to engineered reservoirs with rates $\{\Gamma_\ell\}_{\ell=1}^L$ and range up to $L$ sites. The latter terms correspond to non-local

dissipators with jump operators $\{\hat{a}_m + e^{-\mathrm{i}\theta_\ell}\hat{a}_{m+\ell}\}_{\ell=1}^L$ and relative phases $\{\theta_\ell\}_{\ell=1}^L$; they realize a dissipative analogue of $\hat{\mathcal{H}}_J$ and their range is intended to match the hopping term in Eq. (5). The $L$ distinct engineered reservoirs can be implemented as indirect hopping via an auxiliary, fast decaying, mode or coupling to a transmission line [30]. The gain processes (8), on the other hand, are simply local incoherent pumps with rate $\kappa$, which is assumed to be the same for all cavities; these pumps can be implemented in various ways, e.g. via parametrically coupled auxiliary modes which are then adiabatically eliminated [28].

Since any two cavities which are less than $L$ sites apart are coupled via both photon tunneling and dissipative coupling, interference can build up between these two 'paths', as witnessed by the relative phases $\{\theta_\ell\}_{\ell=1}^L$. These phases are gauge invariant and act like an effective magnetic flux for the photons. They can been implemented in various platforms, e.g. in time-modulated optomechanical systems [65, 66]. In the present model we set all amplitudes $J_\ell$ to be real. One could consider a more general model comprising complex $J_\ell \in \mathbb{C}$ as gauge invariant phases could develop between terms with $\ell \neq \ell'$, without invoking a paired dissipative coupling. We treat this extension in Appendix B, where we show that this case allows for non-reciprocity without opening a point gap in the corresponding spectrum under PBC.

## 4 Non-Hermitian Hamiltonian

Starting from the open quantum system illustrated above, we can obtain a NH Hamiltonian ruling the dynamics of the cavity amplitudes. Using the explicit expressions of Hamiltonian, loss and gain terms, described in Eqs. (4) to (8), the evolution of the mean cavity amplitudes Eq. (3) takes the form

$$\dot{\alpha}_m = -\mathrm{i}\sum_n H_{mn}\alpha_n - \sqrt{\gamma}\alpha_{\mathrm{in},m}\,, \tag{9}$$

where we introduced the non-Hermitian Hamiltonian $H = \sum_{mn}H_{mn}|m\rangle\langle n|$ (here we use the Dirac notation $\{|m\rangle\}$ for the site basis), whose real and imaginary part are given by

$$\mathrm{Re}H_{mn} = \omega_\mathrm{c}\delta_{mn} + \sum_{\ell=1}^L\left[\left(J_\ell - \frac{\Gamma_\ell}{2}\sin\theta_\ell\right)\delta_{m,n-\ell} + \left(J_\ell + \frac{\Gamma_\ell}{2}\sin\theta_\ell\right)\delta_{m-\ell,n}\right], \tag{10}$$

$$\mathrm{Im}H_{mn} = -\gamma_\mathrm{eff}\delta_{mn} - \sum_{\ell=1}^L\frac{\Gamma_\ell}{2}\cos\theta_\ell\left(\delta_{m,n-\ell} + \delta_{m-\ell,n}\right). \tag{11}$$

In Eq. (11) we introduced the total on-site rate of dissipation

$$\gamma_\mathrm{eff} = \frac{1}{2}\left(\gamma - \kappa + 2\sum_{\ell=1}^L\Gamma_\ell\right), \tag{12}$$

which will play an important role in our analysis. If we further move to Fourier space $\alpha_m(\omega) = \int_{-\infty}^{+\infty}\mathrm{d}t\exp(-\mathrm{i}\omega t)\alpha_m(t)$, Eq. (9) takes the simple expression $\alpha(\omega) = -\sqrt{\gamma}\chi(\omega)\alpha_\mathrm{in}(\omega)$, where we grouped the cavity amplitudes and the input fields in the vectors $\alpha = (\alpha_1,\ldots,\alpha_N)^T$ and $\alpha_\mathrm{in} = (\alpha_{\mathrm{in},1},\ldots,\alpha_{\mathrm{in},N})^T$, respectively. The susceptibility matrix (or Green's function) $\chi(\omega)$ describes the spectral response of the system to a given frequency component of the input field, and is given by

$$\chi(\omega) = -\mathrm{i}(\omega\mathbb{1} - H)^{-1}\,. \tag{13}$$

In this way, the properties of NH Hamiltonians can be directly probed in scattering-type experiments. This is made even more explicit by relating the input field to the output field via the input-output relation $\hat{a}_{\text{out},m}(\omega) = \hat{a}_{\text{in},m}(\omega) + \sqrt{\gamma}\alpha_m(\omega)$ [67], to get $\alpha_{\text{out}}(\omega) = S(\omega)\alpha_{\text{in}}(\omega)$, where we introduced the matrix $S(\omega) = \mathbb{1} + \gamma\chi(\omega)$, which is called the scattering matrix of the system [64]. When we probe the system, we drive one site and measure the outgoing amplitude at any of the $N$ output ports. This is the information contained in the scattering matrix, see rightmost column of Fig. 1.

A key feature that can be accessed via $\chi(\omega)$ is non-reciprocity, which occurs whenever $|\chi(\omega)| \neq |\chi(\omega)|^{\mathrm{T}}$ (or equivalently $|S(\omega)| \neq |S(\omega)|^{\mathrm{T}}$) [68, 69]; here the modulus of the matrix is understood as the absolute value of each element. Non-reciprocity entails that the system's response is not invariant upon exchanging the input and the output. In our model, non-reciprocity originates from interference between coherent and dissipative couplings [30]. This is maximal for $\theta_\ell = \frac{\pi}{2}$ ($\theta_\ell = \frac{3\pi}{2}$), giving a total rightward hopping amplitude $J_\ell + \frac{1}{2}\Gamma_\ell$ ($J_\ell - \frac{1}{2}\Gamma_\ell$) and vice versa for the leftward hopping amplitude $J_\ell - \frac{1}{2}\Gamma_\ell$ ($J_\ell + \frac{1}{2}\Gamma_\ell$). Upon further tuning the coupling to the value $J_\ell = \pm\frac{1}{2}\Gamma_\ell$ one can achieve complete suppression of photons travelling in one direction, i.e., unidirectional photon transport [70, 71]. Remarkably, this condition corresponds to an exceptional point (EP) of $H$ of order $N$ [28]; higher-order EPs have recently been at the center of great interest [72, 73]. The opposite case of complete destructive interference corresponds to $\theta_\ell = n\pi$, for which we have fully reciprocal transport.

## 4.1 Open boundary conditions vs. periodic boundary conditions

In Hermitian lattice models the change from OBC to PBC is not abrupt. This reflects the intuition that, for a large enough system, what happens at the boundaries does not affect the bulk properties [3]. In contrast, a change of boundary conditions in a one-dimensional NH lattice with asymmetric hopping is accompanied by the NHSE [13]. In view of this high sensitivity, we need to put special care in the distinction between OBC and PBC. We make this distinction clear below, where we also introduce a convenient re-parametrization of our model in terms of the rescaled hopping constant $\Lambda_\ell = 2J_\ell/\gamma_{\text{eff}}$ and dissipative coupling $\mathcal{C}_\ell = \Gamma_\ell/\gamma_{\text{eff}}$.

The finite-size system of $N$ cavities describes the state of the array under OBC. The NH Hamiltonian in the site basis, given in Eqs. (10) and (11), can be compactly expressed as a Toeplitz matrix

$$H_{mn} = \sum_{\ell=-L}^{L} \mu_\ell \delta_{m,n-\ell}, \tag{14}$$

with coefficients given by

$$\mu_0 = -\delta - \mathrm{i}\gamma_{\text{eff}}, \tag{15}$$

$$\mu_\ell = \frac{\gamma_{\text{eff}}}{2}\left(\Lambda_{|\ell|} - \mathrm{i}\mathcal{C}_{|\ell|}e^{-\mathrm{i}\,\mathrm{sgn}(\ell)\theta_{|\ell|}}\right). \tag{16}$$

For future convenience, we moved to a rotating frame with respect to the frequency $\omega_{\text{d}}$ of the drives (chosen to be monochromatic and with the same frequency) and introduced the detuning $\delta = \omega_{\text{d}} - \omega_{\text{c}}$.

To model the periodic system, we assume PBC, which corresponds to setting $\hat{a}_{N+\ell} = \hat{a}_\ell$ for $\ell = 1, \ldots, L$. Since $H$ describes a translational invariant system, we introduce the plane-wave basis $|k\rangle = \frac{1}{\sqrt{N}}\sum_{m=1}^{N} e^{imk}|m\rangle$, where the quasi-momentum $k$ takes the values $k = 0, \frac{2\pi}{N}, 2\frac{2\pi}{N}, \ldots, 2\pi - \frac{2\pi}{N}$. In this way we obtain a diagonal matrix $H_{kk'} = H(k)\delta_{kk'}$, with eigenvalues

$$H(k) = \sum_{\ell=-L}^{L} \mu_\ell e^{\mathrm{i}k\ell}. \tag{17}$$

The spectrum $H(k)$ describes a single complex-valued energy band [4]. For the case of nearest-neighbor coupling, $L = 1$, we recover the implementation of the Hatano-Nelson model of Eq. (1). In this work, unless stated otherwise, we use $H$ and $H(k)$ to refer to the NH Hamiltonian under OBC and PBC, respectively.

## 5 The quest for trivial topology

The most peculiar feature of the complex spectrum (17) is that $H(k)$ can wind up to $L$ times (both clockwise and counter-clockwise) around the origin. Due to this feature, it is possible to assign a topology directly to the spectrum [5]. To see how this happens, let us first recall the definition of point gap. To avoid confusion, we denote with $\widetilde{H}(k)$ a generic single-band NH Bloch Hamiltonian, not necessarily obtained via the prescription of Sec. 4. $\widetilde{H}(k)$ is said to have a point gap if it describes a curve in the complex plane with an interior. In that case, one can choose a reference point in the interior, called base point, which is gapped from the band, i.e., which does not belong to spectrum [8]. The topology is then assigned to a point-gapped $\widetilde{H}(k)$ via the winding number

$$\nu_{E_b} = \frac{1}{2\pi i} \int_0^{2\pi} dk \left[ \frac{\widetilde{H}'(k)}{\widetilde{H}(k) - E_b} \right],\tag{18}$$

which is defined with respect to the base point $E_b \in \mathbb{C}$. According to this characterization, any point-gapped Hamiltonian is topologically non-trivial [5]. This is because the base point can be chosen arbitrarily, so that it is always possible to take $E_b$ in the interior of the curve. Equivalently, one can describe the situation as the base point being kept fixed and the NH Hamiltonian defined only up to constant complex-energy shifts.

In our framework the situation is strikingly different. In fact, due to the presence of the constant term $\mu_0$, we find that the NH spectrum $H(k)$ is no longer invariant under complex-energy shifts. In particular, a shift along the real axis [see Eq. (10)] corresponds to the off-resonant probing of the system, as set by the detuning $\delta$ (setting $\omega = 0$ in Eq. (13) corresponds to probing on resonance), while a purely imaginary shift [see Eq. (11)] is determined by the total on-site dissipation rate (12) entering the evolution Eq. (9).

In this way, any arbitrariness in the evaluation of the topological invariant is removed. The constant term $\mu_0$ allows to assign a unique value of the winding number to $H(k)$, which is always evaluated *with respect to the origin*. Equivalently, one could remove $\mu_0$ from the complex spectrum (17) and absorb it in the choice of the base point $E_b = -\mu_0 \in \mathbb{C}$, which is then uniquely determined. In the following, we will consider the base point to be the origin and the complex band to retain $\mu_0$. This leads us to the following expression for the NH topological invariant [24, 27, 28]

$$\nu_0 = \frac{1}{2\pi i} \int_0^{2\pi} dk \left[ \frac{H'(k)}{H(k)} \right] \equiv \nu,\tag{19}$$

with the NH Hamiltonian given by Eq. (17) and the coefficients by Eqs. (15) and (16).

A major consequence of our framework is that the properties of $H(k)$ featuring a point gap and $H(k)$ having a non-zero value of the topological invariant are not equivalent; the first is only necessary for non-trivial topology, since clearly there can be point-gapped spectra which do not encircle the origin, see Fig. 1. This has a fundamental implication, which will be explored in Sec. 7: while, according to the topological characterization of effective NH Hamiltonians, non-trivial topology is inescapable for point-gapped spectra, in our framework there is room for point-gapped spectra with trivial topology. This represents a distinctive trait of our framework.

Finally, we comment on the different (but equivalent) characterization of the non-invariance of $H(k)$ under complex shifts given by the parametrization in Eqs. (15) and (16). Consider for simplicity the resonant case $\delta = 0$, i.e., a purely imaginary shift of $H(k)$. In Eqs. (15) and (16), the imaginary shift is effectively reabsorbed by rescaling both the on-site term and the coupling terms by the overall local dissipation rate $\gamma_{\text{eff}}$. In units of $\gamma_{\text{eff}}$, the complex band $H(k)$ is then pinned to $-i$ and the effect of the losses and gains (entering via the rescaled couplings $\Lambda_\ell$ and $\mathcal{C}_\ell$) is to *change the curvature* of $H(k)$, rather than shifting it along the imaginary axis. This is the convention that we will use in all the plots of this work. An example can be seen in Fig. 1, where we show the complex spectrum of our Hatano-Nelson model Eq. (1). In all the plots, unless explicitly mentioned, we will also consider the resonant case $\delta = 0$. We discuss the dependence of the topology on detuning in more detail in Appendix C.

# 6 The opening of a point gap: Non-reciprocity and non-normality

Given that featuring a point gap in the spectrum is a prerequisite for assigning the topology via the winding number, it is natural to ask: what causes $H(k)$ to have a point gap in the first place? Indeed, not all NH spectra of the form (17) display a point gap. We will call *degenerate spectrum* a complex spectrum with no point gap, i.e., a periodic—in general complex-valued—function with no interior, and refer to the transition from a degenerate spectrum to a point-gapped one as the *opening of a point gap*. Note that, according to this definition, a Hermitian Bloch Hamiltonian is a particular case of a degenerate spectrum. Employing the same logic that will be used in Sec. 9 to discuss the BBC, we now show how the opening of a point gap in $H(k)$ affects the corresponding system under OBC. In this way, we are able to link the opening of a point gap with the two following properties: non-reciprocity and non-normality.

For the sake of concreteness, we start with the case $L = 1$. By setting $\mu_{\pm 1} = |\mu_{\pm 1}|e^{i\phi_{\pm 1}}$, we can rewrite Eq. (1) as

$$H(k) = \mu_0 + e^{i\phi^+} \left( |\mu_{-1}|e^{-i(k+\phi^-)} + |\mu_1|e^{i(k+\phi^-)} \right), \tag{20}$$

where we introduced the quantities $\phi^\pm = \frac{\phi_1 \pm \phi_{-1}}{2}$. From this expression it is clear that for $|\mu_1| \neq |\mu_{-1}|$, the point gap is open; and conversely, for $|\mu_1| = |\mu_{-1}| \equiv \mu$, we get the degenerate spectrum

$$H(k) = \mu_0 + 2\mu e^{i\phi^+} \cos\left(k + \phi^-\right), \tag{21}$$

which describes a straight line tilted by $\phi^+$ and offset by $\mu_0$. Furthermore, if we look at the explicit expression of the coefficients (16) (from now on for the case $L = 1$ we set $\mathcal{C}_1 \equiv \mathcal{C}$, $\Lambda_1 \equiv \Lambda$, $\theta_1 \equiv \theta$), we see that $|\mu_{\pm 1}| = \frac{\gamma_{\text{eff}}}{2}(\Lambda^2 + \mathcal{C}^2 \mp \Lambda\mathcal{C}\sin\theta)^{1/2}$, from which it follows that $|\mu_1| = |\mu_{-1}|$ is satisfied when either $\mathcal{C} = 0$ or $\Lambda = 0$, i.e., one of the two coupling vanishes and no interference can build up, or $\theta = n\pi$, with $n$ integer. For a point gap to open, we then need both $\mathcal{C}$ and $\Lambda$ to be non-zero and the flux to be $\theta \neq n\pi$. But this is precisely the condition for non-reciprocal transport that we discussed in Sec. 4. We can arrive at the same conclusion by inspecting directly the definition of non-reciprocity based on the susceptibility matrix, Eq. (13). Indeed, it is clear that $|\chi(\omega)| \neq |\chi(\omega)|^{\text{T}}$ as long as $H$ under OBC is such that $|\mu_1| \neq |\mu_{-1}|$. Therefore, non-reciprocity is the physical mechanism responsible for the opening of a point gap: point-gapped spectra arise from enforcing PBC on non-equilibrium non-reciprocal systems.

The second characterization comes from the distinction between normal and non-normal NH Hamiltonians [74]. Normality refers to the property $[H, H^\dagger] = 0$, and is equivalent to $H$ being diagonalizable by a unitary matrix, since for normal matrices the spectral theorem holds.

Enforcing normality on the NH matrix, Eq. (14), under OBC, leads directly to the condition $|\mu_1| = |\mu_{-1}|$, thus implying the degeneracy of the corresponding PBC spectrum Eq. (21). From a mathematical point of view, point-gapped spectra arise from enforcing PBC on non-normal Hamiltonians. Non-normal Hamiltonians form the subset of NH Hamiltonians for which eigendecomposition may become problematic, e.g. by displaying the NHSE; we will see that a faithful description of non-normal Hamiltonians requires switching to the SVD instead. For $L = 1$, we therefore established the equivalence between the following three concepts: the opening of a point gap in the complex spectrum under PBC, non-normality of the NH Hamiltonian $H$ under OBC, and the presence of non-reciprocity, witnessed by an asymmetry in the susceptibility matrix Eq. (13) (or equivalently in the scattering matrix).

Moving to $L \geq 2$ the scenario becomes considerably richer, as the equivalence between these three concepts is broken. For instance, for Toeplitz matrices (14), the normality condition takes the form $\mu_\ell^* \mu_{\ell'} - \mu_{-\ell} \mu_{-\ell'}^* = 0$, with $0 \leq \ell, \ell' \leq L$. By setting $\mu_\ell = |\mu_\ell| e^{i\phi_\ell}$, we obtain two separate conditions

$$|\mu_\ell||\mu_{\ell'}| = |\mu_{-\ell}||\mu_{-\ell'}|, \tag{22}$$

$$\phi_\ell + \phi_{-\ell} = \phi_{\ell'} + \phi_{-\ell'} \mod 2\pi. \tag{23}$$

The first one contains as a special case $|H| = |H|^{\mathrm{T}}$, while the second condition states that the total phase of any pairs of matrix bands $(-\ell, \ell)$ should be the same. In Fig. 2 we show the opening of a point gap for a model with nearest and next-nearest-neighbor couplings, $L = 2$. By changing the values of the gauge invariant phases $\theta_{1,2}$ from zero to $\pi/2$, the spectrum changes from degenerate (a) to non-degenerate (b), i.e., the point gap opens.

Although for $L \geq 2$ the three characterizations are no longer equivalent, it is still possible to show that a point-gapped spectrum under PBC implies both non-reciprocity and non-normality of the corresponding OBC Hamiltonian. We prove both implications in Appendix A. While the connection between asymmetric hopping and point-gapped spectra has been known for certain models [4], and in fact has been used to provide an explanation of the NHSE [17], a complete understanding of the relationship between point-gapped Hamiltonians and the notion of non-reciprocity has been lacking.

The fact that non-normality and non-reciprocity of $H$ are only necessary for opening a point gap in $H(k)$ implies that: (i) there exist non-reciprocal Hamiltonians under OBC which have a degenerate PBC spectrum. In fact, unlike $L = 1$, for longer range-couplings non-reciprocity can be achieved without opening a point gap. For instance, for complex $\Lambda_\ell$, a gauge invariant phase can be present even in the absence of dissipative couplings, i.e., for all $\mathcal{C}_\ell = 0$, which leads to non-reciprocity without a point gap, see Appendix B. (ii) There exist NH Hamiltonians which are non-normal and yet have a degenerate PBC spectrum. These Hamiltonians feature a kind of non-normality which is not strong enough to lift the degeneracy of $H(k)$, see Appendix B.

# 7 The singular value decomposition and the non-Hermitian gap

After discussing the opening of a point gap, we now want to identify a suitable notion of gap closure for point-gapped spectra. We do this by introducing a new quantity, which we call the *non-Hermitian gap*. We first illustrate the intuitive idea behind it in the top row of Fig. 2, where we show a NH spectrum with $L = 2$, for different values of the dissipative coupling $\mathcal{C}_2$. Starting from a point-gapped spectrum (b) and increasing $\mathcal{C}_2$, $H(k)$ crosses the origin twice, as shown in panels (c) and (d); both crossing events coincide with a change of the winding number, which goes from the initial value $\nu = 0$ to the final value $\nu = 2$. These plots make it clear that the relevant feature that we want to associate with the closure of a gap is captured by the shortest distance from the origin to the spectrum, which vanishes when the complex

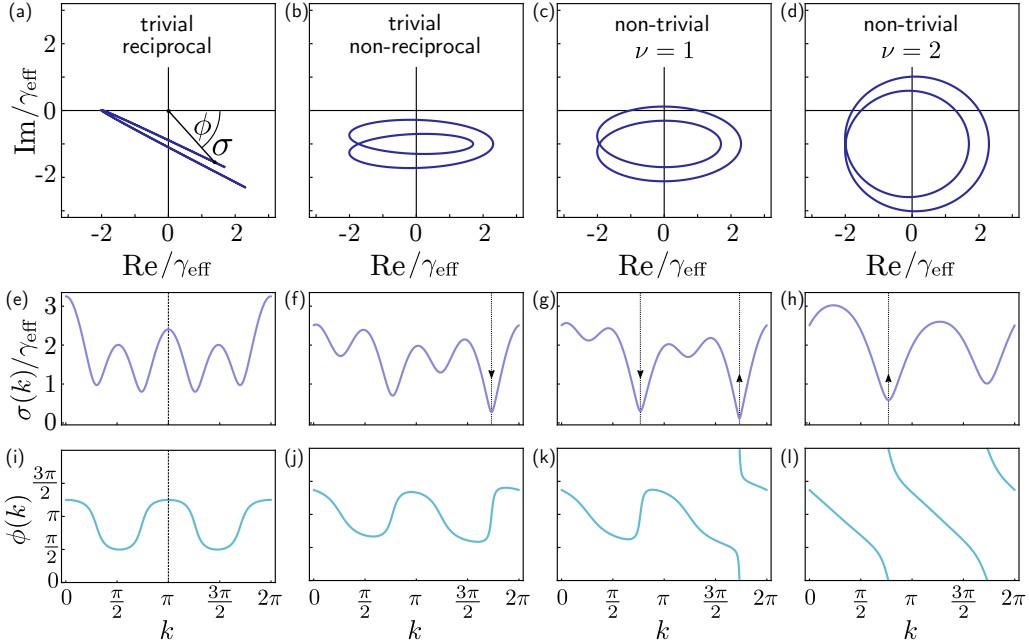

Figure 2: **PBC complex spectrum, singular values and phase.** Plots of the complex spectrum $H(k)$ (a)-(d) and its decomposition, Eq. (27), in terms of singular values spectrum $\sigma(k)$ (e)-(h), and phase $\phi(k)$ (i)-(l), for a model with nearest- and next-nearest-neighbor couplings, $L = 2$. The first column [panels (a), (e) (i)] is for $\theta_1 = \theta_2 = 0$, $\mathcal{C}_2 = 1$, and illustrates a degenerate spectrum; reciprocity is signaled by symmetric $\sigma(k)$ and $\phi(k)$. The other panels illustrate point-gapped (i.e., non-degenerate) spectra for $\theta_1 = \theta_2 = \frac{\pi}{2}$ and increasing values of $\mathcal{C}_2$: $\mathcal{C}_2 = 0.5$ [panels (b), (f), (j)], $\mathcal{C}_2 = 0.9$ [panels (c), (g), (k)], $\mathcal{C}_2 = 1.8$ [panels (d), (h), (l)]. The change of the winding is associated to a topological phase transition, to which corresponds the closing and re-opening of the NH gap, Eq. (28), as indicated by the direction of the arrows in (f)-(h). For all point-gapped spectra, non-reciprocity is witnessed by the asymmetry of $\sigma(k)$ and $\phi(k)$, while a non-trivial winding number, Eq. (19), is accounted for by the number of windings of the phase $\phi(k)$ [panels (k), (l)]. In all panels we set $\mathcal{C}_1 = \Lambda_1 = 0.3$ and $\Lambda_2 = 2$.

spectrum touches the origin. Notice that this happens while the point gap *stays open*, i.e., the spectrum in panels (b), (c) and (d) always encloses a finite area.[1]

The presence of a gain source is instrumental for observing this behavior. In fact, we have seen that opening a point gap necessarily comes with an extra on-site dissipation $\sum_{\ell=1}^{L} \Gamma_\ell$ [see Eq. (12)], coming from the engineered reservoirs. Probing the system further contributes to the on-site loss by an additive factor $\gamma/2$. The reason for introducing a source of gain (8), is precisely to counteract these losses. In terms of the rescaled units employed in Fig. 2, increasing the rate $\kappa$ of gain processes (while keeping the value of all other parameters fixed) corresponds to increasing $\mathcal{C}_2$. The effect of the gain is then to 'inflate' the spectrum, allowing for it to cross the origin. Without gain, the spectrum would be restricted to the lower half of the complex plane, thus preventing any winding around the origin.

We now proceed to formalize the idea illustrated above. To do that, we will use the singular value decomposition (SVD), first introduced in this context in Ref. [24, 25]. For a generic NH

---

[1]Note that our convention differs from that of other works, e.g. Ref. [8], which associate a complex spectrum touching the base point to the point gap closing. In our framework, due to the non-invariance under complex shifts, the distance from the origin naturally defines a gap-closing transition without the need for the spectrum to become degenerate.

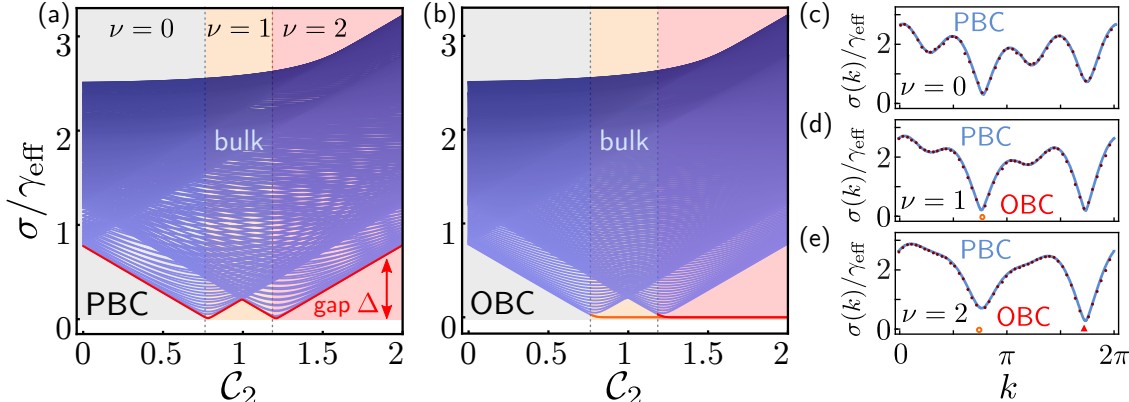

Figure 3: **NH bulk-boundary correspondence and singular values.** Singular value spectrum under PBC (a) and OBC (b) as a function of cooperativity $\mathcal{C}_2$, for a model with nearest- and next-nearest-neighbor couplings, $L = 2$ (same as in Fig. 2). In (a) the non-Hermitian gap, Eq. (28), is shown in red, while in (b) the vanishing singular values in the topological phases are highlighted in red and orange. (c)-(e) PBC and OBC singular value spectra as a function of quasi-momentum in each of the three regimes $\nu = 0, 1, 2$. A non-trivial winding number $\nu$, Eq. (19), leads to the appearance of $|\nu|$ zero singular values under OBC. Here, $\mathcal{C}_1 = \Lambda_1 = 0.3$, $\theta_1 = \theta_2 = \frac{\pi}{2}$, $\Lambda_2 = 2$, $N = 200$.

Hamiltonian $H$, we define the SVD as follows [75]

$$H = U\Sigma V^\dagger = \sum_j \sigma_j |u_j\rangle\langle v_j|, \qquad (24)$$

which decomposes $H$ into the product of a diagonal matrix $\Sigma \equiv \mathrm{diag}(\sigma_1, \ldots, \sigma_N)$ with singular values $\sigma_j \geq 0$ and unitary matrices $U \equiv (|u_1\rangle, \ldots, |u_N\rangle)$, containing the left singular vectors $|u_j\rangle$, and $V \equiv (|v_1\rangle, \ldots, |v_N\rangle)$, containing the right singular vectors $|v_j\rangle$ of $H$. We refer to the set of all singular values as the *singular value spectrum*. The left (right) singular vectors are eigenvectors of the Hermitian product $HH^\dagger$ ($H^\dagger H$) with real and positive eigenvalues $\sigma_j^2$, i.e., they satisfy the following equations

$$HH^\dagger |u_j\rangle = \sigma_j^2 |u_j\rangle, \qquad H^\dagger H |v_j\rangle = \sigma_j^2 |v_j\rangle. \qquad (25)$$

From the definition of the SVD, Eq. (24), it also follows that right and left singular vectors are related to each other in the following way

$$H|v_j\rangle = \sigma_j |u_j\rangle, \qquad H^\dagger |u_j\rangle = \sigma_j |v_j\rangle. \qquad (26)$$

Under PBC, we can express $H(k)$ as product of the distance $|H(k)|$ times the phase $\phi(k) \equiv \mathrm{Arg}\, H(k)$, namely,

$$H(k) = |H(k)| e^{i\phi(k)}. \qquad (27)$$

Since we have $H_{\mathrm{PBC}} = \sum_k H(k)|k\rangle\langle k|$, we readily obtain $\sigma(k) \equiv |H(k)|$, i.e., the singular value spectrum under PBC coincides with the distance of the complex spectrum from the origin. Notice that the identification of these two quantities here is made possible thanks to the fact that the origin provides a fixed reference. For any point-gapped spectrum $H(k)$, we can then define the *non-Hermitian gap* as

$$\Delta = \min_{k \in \mathrm{BZ}} \sigma(k), \qquad (28)$$

which indeed expresses the minimal distance from the complex energy band to the origin.

In the central row of Fig. 2, we show the singular value spectrum and the associated NH gap. We see that $\sigma(k)$ and $\Delta$ closely resemble a single Bloch band and a standard energy gap, respectively. Indeed, when $H$ is Hermitian, $\sigma(k) = |E(k)|$, with sgn $E(k)$ contained in the singular vectors, and $\Delta = \min_{k \in BZ} |E(k)|$, so the two decompositions coincide. From Fig. 2 (f), (g) and (h) we see that the singular value spectrum $\sigma(k)$ is always gapped, except when the topological winding number (19) changes, in which case the NH gap vanishes. This confirms that $\Delta$ properly captures the notion of gap closure and reopening for point-gapped spectra [25], which we take as an evidence of a *NH topological phase transition*. If we look at the bottom row of Fig. 2, we see that the number of windings is accounted for by the phase $\phi(k)$. We stress that, while $\sigma(k)$ can be evaluated for degenerate and point-gapped spectra alike, when defining the NH gap, we restrict ourselves to non-degenerate spectra, i.e., a degenerate spectrum touching the origin should not be associated with the NH gap closing.

Given the insight provided by the singular value spectrum for the case of PBC, we now look at the case of OBC. In Fig. 3, we compare the singular value spectrum under PBC (a) to that under OBC (b), obtained via numerical diagonalization of a finite chain of size $N = 200$. We see that the singular values $\sigma(k)$, plotted as a function of $C_2$, are arranged in a band, where the envelope formed by the smallest singular values determines the NH gap Eq. (28); the NH gap is highlighted in panel (a). Fig. 3 illustrates in a nutshell how, thanks to the SVD, the BBC for NH systems is restored. Indeed, unlike the eigenvalues, the singular values do not suffer any abrupt changes when moving from PBC to OBC: the only difference between the two cases is the emergence of *zero singular values* (ZSVs) [24, 25]—two in the case shown here—under OBC, while the bulk is preserved, see panels (c)-(e). These zero values appear after the closure and reopening of the NH gap, when the system enters a non-trivial phase. The topological invariant constructed from the bulk states correctly counts the number $|\nu|$ of boundary states. To obtain the OBC spectrum in (c)-(e), we write the NH Hamiltonian Eqs. (10) and (11) as the PBC Hamiltonian minus the matrix boundary terms, and express it in the plane-wave basis $|k\rangle$ where the PBC Hamiltonian is diagonal. We then diagonalize the Hamiltonian $\langle k|H_{OBC}|k'\rangle$ and label the eigenstates with $k$. The same approach is used for computing the singular value spectrum in Fig. 1 (third column from the left).

## 7.1 The singular value decomposition and non-reciprocity

We close this section by showing that, beyond witnessing topological transitions, the decomposition (27) encodes extra useful information, as $\sigma(k)$ and $\phi(k)$ allow to diagnose non-reciprocity in full generality. Under PBC, a system is reciprocal if and only if there exists a $k_0$ such that $H(k_0 + k) = H(k_0 - k)$, i.e., up to a constant shift, the PBC spectrum is an even function of $k$. When this condition is fulfilled, the corresponding OBC system is also reciprocal, see Appendix A. Thanks to Eq. (27), this symmetry extends to the singular values $\sigma(k)$ and the phase $\phi(k)$. In Figs. 2 (a), (e), (i) we show the case of a degenerate spectrum that satisfies both $\sigma(\pi + k) = \sigma(\pi - k)$ and $\phi(\pi + k) = \phi(\pi - k)$, and hence is characterized by reciprocal photon transport, also under OBC. In contrast, in Figs. 2 (b), (f), (j) we see that this symmetry is broken for point-gapped spectra. In particular, we note that for topologically non-trivial bands, the phase $\phi(k)$ is always asymmetric due to the winding, see Figs. 2 (k), (l), i.e., non-reciprocity underpins all NH topological phases. In light of the inequivalence between non-reciprocity and point-gapped spectra for $L \geq 1$, see Sec. 6, a degenerate complex spectrum can lead to either reciprocal or non-reciprocal transport under OBC, as shown in Appendix A. In this case, by resolving $\sigma(k)$ and $\phi(k)$ we are still able to detect non-reciprocity, while the same information would not be accessible by inspecting the complex spectrum alone as, for instance, Fig. 2 (a) does not reveal the full dependence of $H(k)$ on $k$.

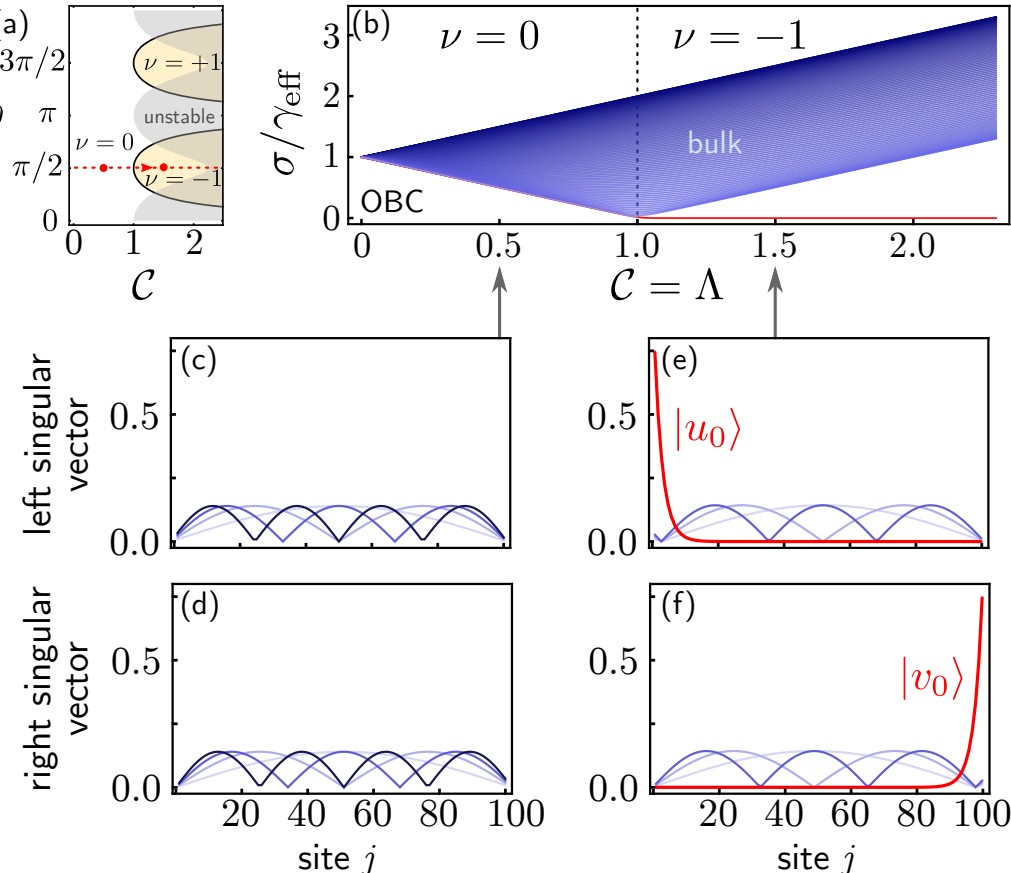

Figure 4: **Bulk-boundary correspondence and the singular vectors for the Hatano-Nelson model.** (a) Topological phase diagram of the Hatano-Nelson model, showing the regions characterized by different values of the topological invariant; the shaded area indicates instability under OBC (cf. Sec. 10) and the path taken in (b) is indicated by the red line. (b) OBC singular values for $\theta = \frac{\pi}{2}$ and at the exceptional point, i.e., $\mathcal{C} = \Lambda$. (c), (e) left and (d), (f) right singular vectors corresponding to the four smallest singular values are shown. The non-trivial phase features a single localized left/right singular vector, corresponding to an exponentially small singular value. We set $\theta = \frac{\pi}{2}$, $N = 100$, (c)-(d) $\mathcal{C} = \Lambda = 0.5$, (e)-(f) $\mathcal{C} = \Lambda = 1.5$.

# 8 Explicit calculation of the zero singular modes

Our results so far suggest that the description of topological phases of NH Hamiltonians (14), once looked through the glass of the SVD, formally resembles that of Hermitian topological insulators: bulk modes are always gapped and ZSVs appear in the NH gap under OBC for non-trivial windings [24, 25]. To further corroborate this picture, we study the behavior of the singular vectors associated with the ZSVs, which we refer to as *zero singular modes* (ZSMs), in analogy to the zero modes of Hermitian systems. Before addressing the general case in the next section, it is instructive to illustrate their properties with a concrete example. Following Ref. [24], we compute explicitly the left and right ZSMs for the case of the Hatano-Nelson model.

We consider the NH Hamiltonian of the Hatano-Nelson model Eq. (1) and we further assume to be at the EP, i.e., $\mu_{+1} = 0$ (the case $\mu_{-1} = 0$ follows analogously). As we discussed in Sec. 4, the EP entails perfect uni-directionality, e.g. only rightward hopping in the case $\mu_{+1} = 0$. The choice of the EP here has the purpose of simplifying the calculations, but simi-

lar conclusions hold also away from the EP. Under PBC, the model has only two regimes: for $|\eta| \equiv |\mu_{-1}/\mu_0| > 1$ the system is topologically non-trivial with winding number $\nu = -1$, while for $|\eta| < 1$ the system is trivial, i.e., $\nu = 0$ [28]. We are interested in the left and right ZSMs $|u_0\rangle$, $|v_0\rangle$, associated with the ZSV $\sigma_0 = 0$, that appear under OBC. By setting $\sigma_0 = 0$ in (26), the two equations decouple and we obtain two independent recurrence relations

$$\mu_{-1} v_0^{(m)} + \mu_0 v_0^{(m+1)} = 0, \qquad \mu_0^* u_0^{(m)} + \mu_{-1}^* u_0^{(m+1)} = 0,$$

in which $v_0^{(m)} \equiv \langle m|v_0\rangle$ ($u_0^{(m)} \equiv \langle m|u_0\rangle$) denotes the $m$th component of the right (left) ZSM, together with the boundary conditions

$$v_0^{(1)} = 0, \qquad u_0^{(N)} = 0, \tag{29}$$

which state that the right (left) ZSM should identically vanish at the left (right) boundary. From these relations we find

$$\frac{v_0^{(m+1)}}{v_0^{(1)}} = \left( \frac{u_0^{(1)}}{u_0^{(m+1)}} \right)^* = \left( -\frac{\mu_{-1}}{\mu_0} \right)^m \equiv (-\eta)^m. \tag{30}$$

We see that, for $|\eta| > 1$, $v_0^{(m)}$ increases exponentially with $m$, while $u_0^{(m)}$ decreases and vice versa for $|\eta| < 1$. It follows that the normalized right (left) singular vectors satisfying Eqs. (29) are given by

$$|v_0\rangle = \mathcal{N} \sum_{m=1}^{N} (-\eta)^{m-1} |m\rangle, \tag{31}$$

$$|u_0\rangle = \mathcal{N} \sum_{m=1}^{N} (-\eta^*)^{N-m} |m\rangle, \tag{32}$$

with $\mathcal{N} \equiv \left( \frac{1-|\eta|^2}{1-|\eta|^{2N}} \right)^{1/2}$. For finite $N$, these solutions do not satisfy the boundary conditions laid out in Eqs. (29). This should not surprise us, as we are demanding that the smallest singular value $\sigma_0$ is exactly zero. The boundary conditions can still be satisfied in the thermodynamic limit, $N \to \infty$, for which $\sigma_0$ exponentially approaches zero. In particular, $v_0^{(1)} = u_0^{(N)} = \mathcal{N} \to 0$ for $N \to \infty$ *only* when $|\eta| > 1$. Since the condition $|\eta| > 1$ is also the requirement for a non-trivial winding number, the ZSM *only* exists in the case of non-trivial NH topology.

This calculation explicitly shows the restored BBC: when $\nu = -1$, there is exactly one right (left) ZSM exponentially localized at the right (left) boundary of the system. On the other hand, for $\nu = 0$, there are no localized states under OBC.

We display the behavior of the (absolute value of the) singular vectors in Fig. 4. Moving through the topological phase diagram (a) of the Hatano-Nelson model at the EP ($\mathcal{C} = \Lambda$), the singular value spectrum under OBC changes, as shown in Fig. 4 (b). In the topologically trivial phase ($\nu = 0$), which corresponds to $\mathcal{C} < 1$ [28], both the left (c) and right (d) singular vectors are extended plane-wave modes. As the NH gap closes ($\mathcal{C} = 1$) and we enter the non-trivial phase for $\mathcal{C} > 1$, an exponentially small singular value appears under OBC that corresponds to a pair of exponentially localized right and left singular vectors, Eqs. (31) and (32), while all other singular vectors remain plane waves, i.e., the bulk is left intact by the change in the boundary term. Left (e) and right (f) singular vectors localize at opposite ends.

## 9 Mapping to the generalized SSH model

The example above illustrates the BBC for the Hatano-Nelson model by finding explicit solutions for both PBC and OBC and putting the two in correspondence. We now show that the

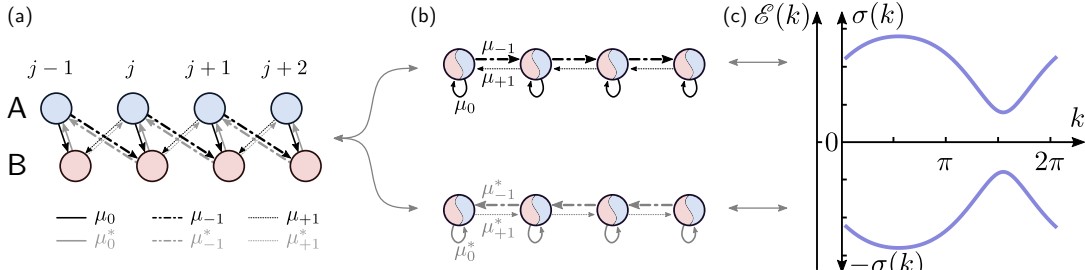

Figure 5: **Mapping between a generalized SSH model and non-Hermitian model for $L = 1$.** A generalized SSH (GSSH) model (a) with sub-lattices $A$ and $B$, intra-cellular hopping amplitude $\mu_0$ and two inter-cellular hopping amplitudes $\mu_1$, $\mu_{-1}$ is equivalent to two copies of the Hatano-Nelson model (b), one with rightward (leftward) hopping $\mu_{-1}$ ($\mu_{+1}$) and the other with rightward (leftward) hopping $\mu_{+1}^*$ ($\mu_{-1}^*$). The two copies are characterized by opposite chirality. (c) Each of the two bands in the spectrum $\mathscr{E}(k)$ of the GSSH corresponds to the singular value spectrum $\pm\sigma(k)$ of the Hatano-Nelson models. The mapping also holds for $L > 1$.

same correspondence holds in general for single-band NH models without symmetry. We do so by means of a mapping between the SVD of a NH lattice and the eigendecomposition of a generalized Su-Schrieffer-Heeger (GSSH) model in a doubled space.

From Eq. (26) it is easy to see that the SVD (25) can be equivalently obtained from the eigenvalues and eigenvectors of the doubled matrix $\mathscr{H}$ [5]

$$\mathscr{H} \equiv \begin{pmatrix} 0 & H^\dagger \\ H & 0 \end{pmatrix} = |B\rangle\langle A| \otimes H + |A\rangle\langle B| \otimes H^\dagger, \qquad (33)$$

in which we introduced basis vectors labelled $A$ and $B$. The same construction appeared in Refs. [5, 24, 25, 76]. In particular, for the special case $L = 1$, Ref. [24] made use of the same mapping to address the link between Hermitian topological insulator theory and amplification in driven-dissipative arrays. To prove the BBC for arbitrary integer values below, it is essential that we use the *generalized* Hatano-Nelson model with $L > 1$ and map it to the *generalized* SSH model. $\mathscr{H}$ is Hermitian by construction and is endowed with a sub-lattice structure. This readily reveals the connection with the Hermitian SSH model [77, 78]. In fact, $\mathscr{H}$ is mathematically equivalent to the Hamiltonian of a GSSH model, i.e., a Hermitian bipartite lattice with long-range hopping between different sub-lattices up to a range $L$ [79, 80]. Under OBC, we can rewrite Eq. (33) explicitly as

$$\mathscr{H} = \sum_{j=1}^{N} \mu_0 |B, j\rangle\langle A, j| + \sum_{\ell=1}^{L}\sum_{j=1}^{N-\ell} \mu_\ell |B, j\rangle\langle A, j+\ell| + \sum_{\ell=1}^{L}\sum_{j=\ell+1}^{N} \mu_{-\ell} |B, j\rangle\langle A, j-\ell| + \text{H.c.} \qquad (34)$$

From the above expression we identify $\mu_0$ as the intra-cellular hopping amplitude between sub-lattices $A$ and $B$, and $\mu_\ell$, $\mu_{-\ell}$ as the inter-cellular hopping amplitudes [78], as we show in Fig. 5 for the case $L = 1$. In particular, $\mu_{\ell<0}$ couples sub-lattice $A$ to sub-lattice $B$ of the $\ell$th neighbor to the right, while $\mu_{\ell>0}$ couples sub-lattice $B$ to sub-lattice $A$ of the $\ell$th neighbor to the left. The GSSH model differs from the standard SSH model by the presence of these *two* different types of inter-cellular hopping amplitudes. Remarkably, the term $\mu_0$, which was originally derived [see Eq. (15)] to keep track of the real (imaginary) shift of $H(k)$ due to the probe frequency (fluctuation-dissipation processes), now plays the role of intra-cellular hopping. The nonequivalence between shifted complex spectra, which characterizes our framework, is then essential to establish the mapping to the GSSH model.

Under PBC, the GSSH Hamiltonian takes the form

$$\mathcal{H}(k) \equiv \begin{pmatrix} 0 & H^*(k) \\ H(k) & 0 \end{pmatrix}$$

$$= \begin{pmatrix} 0 & \sum_{\ell=-L}^{L} \mu_\ell^* e^{-ik\ell} \\ \sum_{\ell=-L}^{L} \mu_\ell e^{ik\ell} & 0 \end{pmatrix}, \tag{35}$$

from which we can make the following observations. First, the winding number of the GSSH model coincides with the NH winding number of $H(k)$, Eq. (19). Second, the eigenvalues of $\mathcal{H}(k)$, which define the energy bands of the GSSH model, are given by $\mathcal{E}(k) = \pm\sigma(k)$, i.e., correspond to the singular value spectrum of $H(k)$, Fig. 5 (c). From this it also follows that (i) gap closing transitions of the GSSH model coincide with the closing of the NH gap, (ii) ZSMs are zero-energy (mid-gap) modes of the GSSH model.

The eigenvectors of $\mathcal{H}(k)$ are given by

$$|\psi_-(k)\rangle = \frac{1}{\sqrt{2}} \begin{pmatrix} -e^{-i\phi(k)} \\ 1 \end{pmatrix}, \quad |\psi_+(k)\rangle = \frac{1}{\sqrt{2}} \begin{pmatrix} e^{-i\phi(k)} \\ 1 \end{pmatrix}, \tag{36}$$

with the phase $\phi(k) \equiv \text{Arg}H(k)$ of Eq. (27) and $+$ $(-)$ stands for the positive (negative) eigenvalue of $\mathcal{H}(k)$. The singular vectors of $H(k)$ corresponding to the singular values $\sigma(k)$ can then be expressed as

$$|A, v(k)\rangle = \frac{|\psi_+(k)\rangle + |\psi_-(k)\rangle}{\sqrt{2}}, \tag{37}$$

$$|B, u(k)\rangle = \frac{|\psi_+(k)\rangle - |\psi_-(k)\rangle}{\sqrt{2}}. \tag{38}$$

The singular vectors also encode the topological invariant which we obtain as difference between the Zak phases of right and left singular vector [78, 81]

$$\psi_{\text{Zak}}^{\pm} = -i \int_0^{2\pi} dk \, (\langle \psi_+(k)|\partial_k \psi_+(k)\rangle - \langle \psi_-(k)|\partial_k \psi_-(k)\rangle)$$

$$= -i \int_0^{2\pi} dk \, (\langle v(k)|\partial_k v(k)\rangle - \langle u(k)|\partial_k u(k)\rangle) = \pi \nu. \tag{39}$$

This corresponds to taking the difference between Zak phases of different bands in the Hermitian SSH model which is gauge invariant.

Moving to OBC, we obtain the singular vectors as eigenvectors of Eq. (34). The mapping from a one-band non-Hermitian model to a Hermitian SSH model in a doubled space allows us to import the Hermitian BBC for the GSSH model; for a rigorous proof see Ref. [79]. In turn, thanks to the equivalence between the topological invariants of the two models and to the SVD, this mapping restores the BBC for NH systems. The winding number $\nu$ as defined on $H(k)$ corresponds to $|\nu|$ localized eigenvectors of $\mathcal{H}$ under OBCs which, via the SVD, implies $|\nu|$ localized singular vectors of $H$. The corresponding singular values are zero in the thermodynamic limit $N \to \infty$ (exponentially small singular values $\log \sigma \propto -N$). Left (right) singular vectors correspond to the eigenvector contributions on the $B$ ($A$) sub-lattices of the GSSH model so the left and right singular vectors localize at opposite ends. This proves the statement of the NH BBC made in the introduction.

## 10 Non-Hermitian topological amplification

We are now ready to explore more in depth the physical consequences of the re-established BBC. Due to the driven-dissipative nature of the systems involved, we expect the properties

(a)
(b)

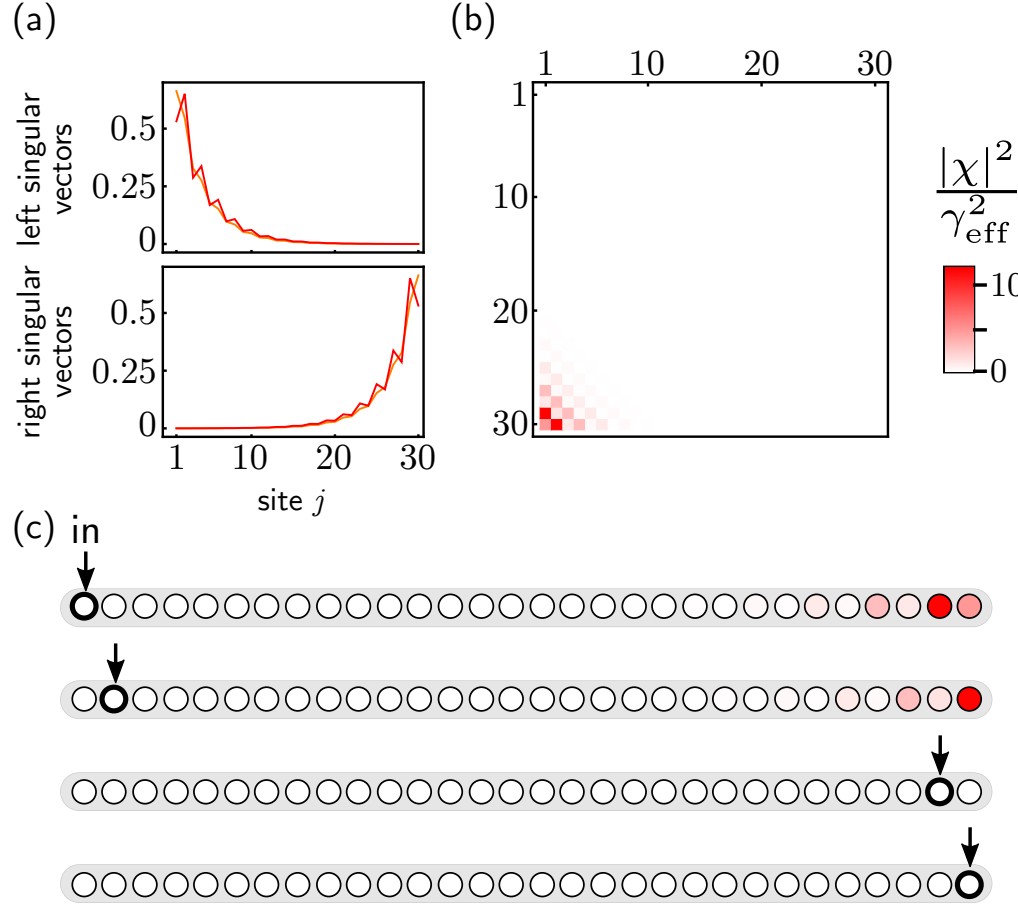

$\dfrac{|\chi|^2}{\gamma_{\text{eff}}^2}$

(c)

Figure 6: **Zero singular modes and topological amplification.** (a) A winding number of $\nu = -2$ leads to a pair of two singular zero modes which (b) dominate the susceptibility matrix (41). Note that the diagonal entries are of order 1, although not discernible from the plot. (c) This corresponds to two channels of directional amplification as can be seen from the steady-state amplitudes. Here, driving site one yields the highest gain at site $(N-1)$ and driving site two yields the highest gain at site $N$ while the reverse gain is strongly suppressed. Here, $N = 30$, $\mathcal{C}_1 = \Lambda_1 = 0.05$, $\theta_1 = \theta_2 = \frac{\pi}{2}$, $\mathcal{C}_2 = 1.9$, $\Lambda_2 = 2$.

of NH topological phases to be completely different with respect to those of standard (Hermitian) topological phases. In particular, we draw a connection to the physical interpretation of the ZSMs [24, 28], which are the relevant edge modes entering the correspondence. Due to the lack of a BBC for point-gapped spectra, the physical properties of NH topological edge states under OBC have remained elusive, with investigations mostly focusing on semi-infinite boundary conditions [5, 16, 17, 56].

We have already established that behind any point-gapped spectrum lies non-reciprocity, and that transitions to a non-trivial topological phase further require the presence of (sufficiently large) gain. We therefore anticipate that the transport properties of the system will be related to these two ingredients. Some works on bosonic implementations of NH lattices have also revealed the presence of directional transport together with an amplification mechanism [24, 28, 57, 82–84]. In particular, Ref. [24] first employed the decomposition of the susceptibility matrix in terms of the SDV to study topological amplification (for $L = 1$) at the level of $\mathcal{H}$, while in Ref. [28] we provided (for $L = 1$) a precise correspondence between non-trivial NH topology and directional amplification, which motivates the present work.

We now look at the system under OBC and relate the results of the BBC to the system's transport properties. We do so by expressing the susceptibility matrix under OBC, Eq. (13), in terms of the SVD, Eq. (24). We expand the susceptibility matrix $\chi(\omega)$, Eq. (13), for finite $N$ in terms of the singular values and vectors of $(\omega\mathbb{1} - H) = \sum_j \sigma_j |u_j\rangle\langle v_j|$ [24] (for brevity, we omit the argument $\omega$ from the singular values and vectors)

$$\chi(\omega) \equiv -\mathrm{i}(\omega\mathbb{1} - H)^{-1} = -\mathrm{i}\sum_j \frac{1}{\sigma_j}|v_j\rangle\langle u_j|. \qquad (40)$$

From the BBC, we know that, when the topology is non-trivial ($\nu \neq 0$), there are $|\nu|$ exponentially small singular values $\sigma_j$, approximately separated from the bulk by the NH gap (28), to which correspond $|\nu|$ ZSMs. In this case, the main contribution to $\chi(\omega)$ comes from the ZSMs and the bulk modes can be neglected, namely [24],

$$\chi(\omega) \cong -\mathrm{i}\sum_{j\in\mathrm{ZSMs}} \frac{1}{\sigma_j}|v_j\rangle\langle u_j| \qquad (\nu \neq 0). \qquad (41)$$

This expression provides a direct insight into the physical meaning of nontrivial topology by noting the following. First, since the $\sigma_j$ are exponentially small, the contribution $1/\sigma_j$ leads to an exponentially large multiplication factor, which characterizes the response of the system to an input probe, see Sec. 3; the element $|\chi(\omega)|^2_{m,n} > 1$ can in fact be associated with a gain factor [24, 28]. Second, the left and right ZSMs $|u_j\rangle$, $|v_j\rangle$ are exponentially localized *at opposite ends*, which leads to end-to-end non-reciprocal transport. The left singular vectors select the input site and the right singular vectors the output site with the largest gain. We can see this explicitly by inspecting the case of $L = 1$, where we have $\chi(\omega)_{m,n} \propto \sigma_0^{-1} v_0^{(m)}[u_0^{(n)}]^*$ with the analytic expressions of the exponentially localized right and left ZSMs given in Eqs. (31) and (32) [24]. Taken together, these two features imply that the off-diagonal corners dominate the matrix (41). Physically, this corresponds to exponential amplification of a weak coherent signal in one direction and exponential attenuation in the reverse direction. This unique scaling of the gain accompanying directional amplification is the hallmark of NH topological phases under OBC, which we refer to as *non-Hermitian topological amplification* [24, 28]. In a topologically non-trivial phase, thanks to Eq. (41), the response to a probe field is proportional to the right ZSM $\alpha(\omega) = -\sqrt{\gamma}\chi(\omega)\alpha_{\mathrm{in}}(\omega) \propto v_0$, which provides a clear physical interpretation of the ZSM.

For higher winding numbers, $|\nu| > 1$, we obtain multiple *linearly independent* ZSMs, and hence $|\nu|$ channels for directional amplification, as we show in Fig. 6 for $L = 2$, with the sign of $\nu$ selecting the direction of the amplification. NH topological amplification entails that the ZSMs are directly measurable in a simple transmission experiment and the topological winding number Eq. (19) can be extracted by counting the number and direction of amplified edge modes, without having to measure the momentum-resolved complex energy band [85]. Furthermore, amplification is only possible thanks to the bosonic nature of our implementation, as for fermionic systems the pile-up of excitations at the boundary would be forbidden by the exclusion principle.

The susceptibility matrix is only meaningful if the system converges to a well-defined steady state, i.e., if the system is dynamically stable, in which case the steady-state cavity amplitudes can be expressed in terms of the susceptibility matrix $\alpha_{\mathrm{ss}} = \mathrm{i}\sqrt{\gamma}H^{-1}\alpha_{\mathrm{in}} = \sqrt{\gamma}\chi(0)\alpha_{\mathrm{in}}$. Dynamical stability is governed by the imaginary parts of the eigenvalues $\lambda_m$, with $\mathrm{Im}\,\lambda_m < 0$ for all $m$ indicating decay to a steady state [24, 28]. For point-gapped Hamiltonians, the OBC spectrum differs drastically from the PBC spectrum—another feature typically associated with the NHSE (see Fig. 1). Therefore, the eigendecomposition, rather than revealing the BBC, plays the key role of determining the dynamical stability of the OBC system. While non-trivial

topology under PBC *requires* negative and positive imaginary parts (instability) of the eigenvalues so that $H(k)$ encircles the origin, the OBC spectrum can lead to dynamical *stability* (see Fig. 1). Under PBC, non-trivial topology requires non-reciprocity to open the point gap, and unavoidably instability for some $k$. This implies directionally propagating cavity fields that grow exponentially in time as they revolve around the cavity ring. Moving to OBC interrupts this motion, which can make the system stable, and leads to the directional pile-up of excitations at one end of the chain which can be extracted as directional amplification. This is reflected in the system response to a resonant probe as encoded in the susceptibility matrix, Eq. (41). In the literature e.g. on non-equilibrium pattern formation [86], solitons [87], and lasing [88], this situation is called a convective instability and is distinguished from an absolute instability which in our system corresponds to the case $\max_m \text{Im}\lambda_m > 0$. Such convective type of instability is the physical mechanism through which excitations can leave the system under open boundary conditions, which can stabilize the NH topological phase. In the future, we will explore consequences of our results presented here in the presence of interactions, e.g. in arrays of nonlinear resonators, which allow for a richer hydrodynamic description [89]. To be explicit, dynamic stability *always* at the very least requires local decay, i.e. $\text{Im}\mu_0 < 0$, although the requirement typically has to be even stricter. For instance, for $L = 1$, the $m$th eigenvalue, is given by [24, 28]

$$\lambda_m = -\delta - \text{i}\frac{\gamma_{\text{eff}}}{2}\left[1 - \sqrt{\mathcal{C}^2 - \Lambda^2 + 2\text{i}\mathcal{C}\Lambda\cos\theta}\cos\left(\frac{m\pi}{N+1}\right)\right],\tag{42}$$

so here dynamic stability further requires the real part of the square-root term to be smaller than one.

## 11 Robustness against disorder

Among the defining properties of topology is robustness against disorder. Given that NH topology exists in the absence of symmetries, it is not immediately clear whether NH topological phases enjoy such robustness, and, if so, where it stems from. As we show here, the fact that the susceptibility matrix $\chi(\omega)$ is dominated by the ZSMs, Eq. (41), separated from the bulk modes by the gap $\Delta$, guarantees the robustness of NH topological phases against disorder [90].

For concreteness, we consider both a non-trivial and trivial NH Hamiltonian (14) for $L = 1$ (with the same parameters as in Fig. 1) to which we add a local, disordered, complex potential $\xi_j$, such that the new NH Hamiltonian is given by $H - \text{i}\,\text{diag}(\xi_1,\ldots,\xi_N)$. We numerically sample multiple realizations assuming disorder of the decay rates with compact support, i.e., $\xi_j \in [-w, w]$, and compute a histogram of the singular values under OBC as a function of $k$ as well as the (left) singular vectors of a few representative realizations. We show the results for moderate disorder in Fig. 7.

In both the initially non-trivial (a) and trivial (c) case, the singular values distribute around the disorderless $\sigma(k)$ (dashed black curve) and their distribution is only slightly deformed at the extrema. The main feature is that the ZSM present in the disorderless non-trivial case persists in *all* disordered realizations, see Fig. 7 (a), so the susceptibility matrix is still dominated by the ZSM, which remains localized and is hardly affected by the disorder, see (b). The bulk modes, on the other hand, start to localize, as we would expect for a one-dimensional model due to Anderson localization [91]. However, since these are separated from the ZSM by the NH gap, their contribution is negligible. We therefore conclude that NH topological amplification is robust against disorder as long as the NH gap does not close, i.e. the tolerable disorder strength is set by the NH gap. For the standard case of disorder with compact support, i.e., $|\xi_j| \le w$, we obtain a simple sufficient criterion for robustness: the non-trivial phase is robust

to the presence of disorder whenever

$$\Delta > \gamma_{\text{eff}} w. \tag{43}$$

As shown by the shaded area in the inset of Fig. 7 (a), the effect of disorder on the PBC spectrum is at most a shift by $\gamma_{\text{eff}} w\, e^{i\zeta}$ with some phase $\zeta \in (-\pi, \pi]$, which is achieved when *all* sites independently saturate the bound, i.e., $|\xi_j| = w$ for all $j = 1, \dots, N$. As long as the inner bound imposed on the PBC spectrum by this maximally disordered configuration does not cross the origin, robustness is guaranteed [90]. On the other hand, when Eq. (43) is not fulfilled, the disorder may induce a transition to a topological trivial phase (depending on the specific realization at hand) and robustness is no longer guaranteed. In contrast, in the initially trivial case, there is no ZSM so the susceptibility matrix is fully determined by the bulk modes, which we observe to localize, see Fig. 7 (d). We expect our results to extend also to more general types of disorder [90].

Finally, we notice that topological robustness can also be addressed by exploiting the mapping to the GSSH of Sec. 9. The GSSH enjoys chiral symmetry $(\sigma_z \otimes \mathbb{1}_N)\mathcal{H}(\sigma_z \otimes \mathbb{1}_N) = -\mathcal{H}$ which ensures the presence of the gap and then ensures that (Hermitian) topology is robust to perturbations that do not break this symmetry [78]. Disordered NH models still map to the GSSH model for *any* type of disorder so the chiral symmetry of the associated Hermitian model is always preserved. This allows to infer the robustness of NH topological phases for each of the two NH copies. Here, to highlight the self-consistency of our framework and the role of the NH gap, we discussed the robustness to disorder fully at the NH level.

## 12 The non-Hermitian skin effect is not topological

Since we saw that the SVD allows to restore the BBC, we can now turn our attention to the eigenvectors. In point-gapped systems, a macroscopic number of the eigenvectors localizes at one edge of the system due to the NHSE [9, 13]. We established that the set of topologically non-trivial systems presents only a sub-set of those with point gap and NHSE (see Fig. 1), which rules out the topological origin of the NHSE [16, 17]. This result relies on (i) the non-equivalence of point-gapped spectra under complex shifts and (ii) working with the SVD. These ingredients go hand in hand since the SVD, unlike the eigendecomposition, is not invariant under shifts. In fact, for non-normal matrices, singular values and vectors can change non-trivially due to a diagonal shift. Relinquishing one of the two points, as in Ref. [25] for (i), does not reveal the full extent of the BBC for point gapped Hamiltonians. Relinquishing both points, i.e., assuming arbitrary base points and using the eigendecomposition, in general obscures the nature of the BBC for point-gapped spectra due to the NHSE.

Aside from NH topology, we can still ask what physical role is played by the NHSE. For $L = 1$, an open point gap under PBC coincides with non-reciprocity and non-normality under OBC, see Sec. 6. In this case, the ensuing NHSE can be given a clear physical interpretation, as it is put in one-to-one correspondence with non-reciprocal photon transport without gain. This is clearly visible in Fig. 1, by comparing the behavior of the eigenvectors and the susceptibility matrix.

For $L \geq 2$, the point gap opens through the interplay of non-normality [Eqs. (22)-(23)] and non-reciprocity, with neither of them alone providing a sufficient condition for the NHSE, as we show in Appendix A. Non-normality implies that the eigenvectors become linearly dependent and left and right eigenvectors differ, such that $H$ can no longer be unitarily diagonalized. Since the localized eigenvectors are clearly not linearly independent, non-normality is a necessary (but not sufficient) condition for the NHSE [4]. In Appendix B, we provide examples of both non-normal but reciprocal, and non-reciprocal but normal NH Hamiltonians, which

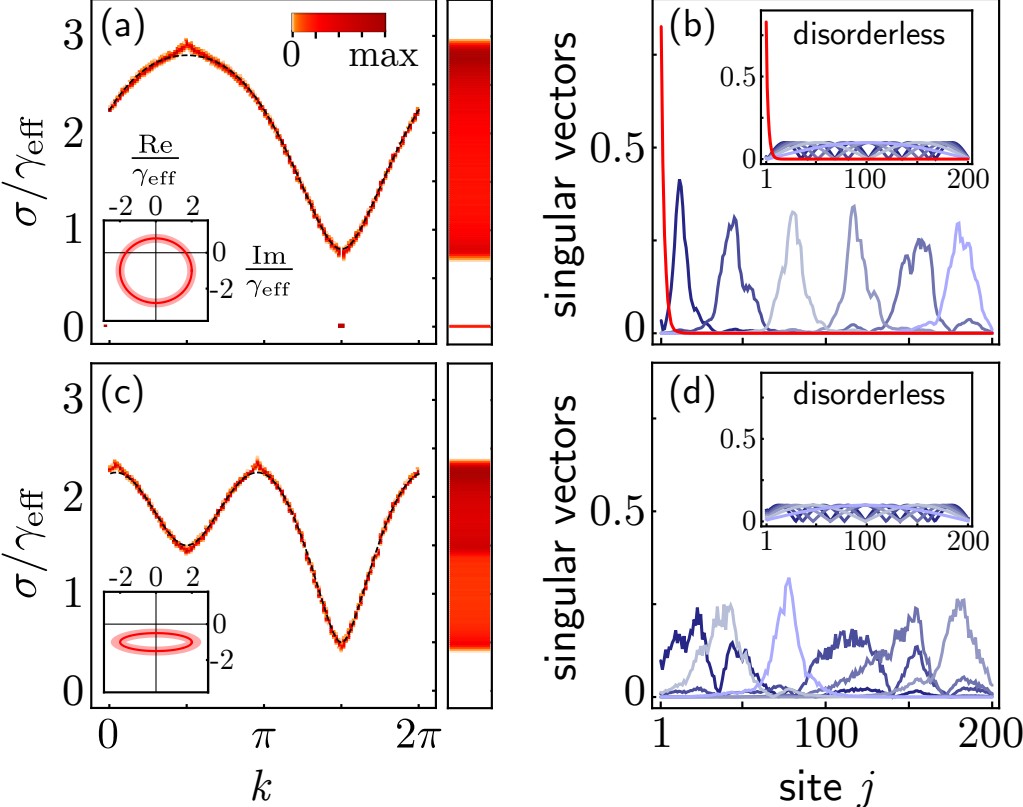

Figure 7: **Robustness of topological amplification against disorder in the Hatano-Nelson model.** (a), (c) Histogram of the singular values under OBC as a function of $k$ and (inset) theoretical bounds of the PBC spectrum according to Ref. [90]. The dashed black curve indicates the singular value spectrum in the absence of disorder. (b), (d) Some (left) singular vectors of a representative realization which localize due to disorder and (inset) corresponding disorderless right singular vectors. Here, $\Lambda = 2$, $\theta = \frac{\pi}{2}$, $w/\gamma_{\text{eff}} = 0.25$, (a)-(b) $\mathcal{C} = 1.8$, (c)-(d) $\mathcal{C} = 0.5$.

do not show exponentially localized eigenvectors. We therefore conclude that the NHSE is the result, visible under OBC, of non-normality and non-reciprocity acting jointly. Only if we restrict to point-gapped Hamiltonians the NHSE becomes a proxy for non-reciprocal transport, thus acquiring a clear physical interpretation.

In conclusion, the NHSE detects interesting matrix anomalies, but their dramatic effects are not necessarily linked to physically observable consequences. This is further supported by the fact that the NHSE is neither directly observable in the scattering matrix nor is present in the steady-state fluctuations [29]. We suggest to use the singular value decomposition (SVD) for point-gapped Hamiltonians instead, which is the appropriate tool to analyze steady states and scattering matrices or Green's functions.

# 13 Conclusion and Outlook

Our study provides an alternative route to the classification of non-Hermitian (NH) topological phases, where the focus is shifted from effective NH Hamiltonians to NH models implemented in driven-dissipative arrays of cavities and *probed* via standard transmission measurements [24, 28]. This pragmatic approach results in the fact that point-gapped spectra differing

by a complex-energy shift are not equivalent or, in other words, fixes the origin of the complex plane as the common reference (base point) for the evaluation of the topological winding number. We showed that this seemingly innocuous change has in fact deep consequences for the topological characterization of NH Hamiltonians. By means of the singular value decomposition [24,25], we introduced alternative quantities (listed in parenthesis) to respectively characterize the bandstructure (singular value spectrum), gapped phases and gap-closing topological phase transitions (NH gap) and topological zero modes (zero singular modes) of NH Hamiltonians. In terms of these quantities, we formulated and proved a bulk-boundary correspondence (BBC) for NH point-gapped systems, which is one of the outstanding problems in the field of non-Hermitian topology [4]. The framework we presented is self-contained and our main results can be found summarized in Fig. 1, for the realization of the Hatano-Nelson model [14].

Our results should be regarded as alternative to—and not in conflict with—the current topological characterization of effective NH Hamiltonians [4,5], since the two approaches are built on different assumptions. However, some conceptual aspects that appear challenging—or even paradoxical—within the current approach based on effective NH Hamiltonians, become instead particularly simple in our framework. Here we stress three such aspects: first, changes from periodic to open boundary conditions need not to be associated with a topological phase transition [16], i.e., there is no need to extend the notion of point-gap topology to finite-size systems; as in standard topological band theory, topology pertains the bulk system and the BBC provides a bridge to observable effects at the boundaries [1]. Second, by replacing the eigendecomposition with the singular value decomposition, we avoid to introduce any modifications to the Bloch band theory [13,18–20]. Third, NH topological phases correspond to steady-state phenomena directly observable in finite-size systems, in contrast to the transient dynamics of semi-infinite systems [5,56] or to non-unitary quantum dynamics relying on post-selection [92], and without the need for bulk probes [93].

Our framework is by design especially suited for photonic implementations. In particular, the neat connection between NH topology and the system's scattering response enables the experimental validation of the NH BBC. In this respect, optomechanical and photonic platforms are ideal candidates to implement NH topological amplification. Specifically, in superconducting circuit optomechanics, directional amplification has been realized in few-mode systems [63,94] and the control of multi-mode arrays recently demonstrated [52]. Nano-optomechanical systems [50,95] as well as multiscale optomechanical crystal structures [51] are also ideal platforms where to implement NH lattice dynamics. Other systems, such as coupled waveguide arrays [37,96,97], exciton-polariton microcavities [45,46] and topolectric circuits [98,99] are also excellent candidate platforms. Our work naturally opens the door to the study of other sources of gain, such as parametric processes [100,101], with applications to the design of novel lattice amplifiers and sensors [57–59].

On the theoretical side, exciting lines for future enquiries include extending our approach to NH models with multiple bands and in higher dimensions. Multi-band models can endow the NH Bloch Hamiltonian with symmetries, for which a classification in terms of 38 symmetry classes was recently proposed [5,8]. The impact of these symmetries on the system's transport properties has been unexplored so far, which would be an ideal task for our framework. Multi-band, one-dimensional lattices are also the simplest setting in which both non-trivial Hermitian and NH topology can co-exist as well as point and a line gaps, thus allowing to study their interplay. Furthermore, it is worthwhile exploring whether a BBC based on the singular value decomposition generalizes to higher dimensions. Higher dimensions bring up new questions, such as the definition of meaningful topological invariants [8,102] and new possibilities, such as a non-Hermitian topological insulator embedded in a three dimensional system [103] and a reciprocal skin effect [104].

## Acknowledgments

**Funding information** M.B. acknowledges funding from the Swiss National Science Foundation (PCEFP2_194268). C.C.W. acknowledges the funding received from the Winton Programme for the Physics of Sustainability and EPSRC (EP/R513180/1). This work is supported by the European Union's Horizon 2020 research and innovation programme under grant agreement No 732894 (FET-Proactive HOT).

## A  Relation between non-reciprocity, non-normality and the singular value decomposition

In this Appendix we show that, for $L \geq 2$, the concepts of point-gapped spectrum under PBC, non-normality, and non-reciprocity under OBC are no longer equivalent, see Fig. 8. As explained in Sec. 7.1, the asymmetry of the SVD under PBC is equivalent to non-reciprocity under OBC. We first prove this statement, and then show that the opening of a point gap implies both non-normality and non-reciprocity.

### A.1  The singular value decomposition and non-reciprocity

First, we prove the claim of Sec. 7.1, namely that non-reciprocity under OBC corresponds to at least one of the two real-valued functions $\sigma(k)$, $\phi(k)$ in Eq. (27) not being an even function in quasi-momentum space. This means that for any $k_0$ in the Brillouin zone, we have

$$
\begin{aligned}
\phi(k_0 + k) &\neq \phi(k_0 - k), \\
\text{or } \sigma(k_0 + k) &\neq \sigma(k_0 - k).
\end{aligned}
\tag{A.1}
$$

As an illustrative case, consider the Hatano-Nelson model at the EP, for which the PBC spectrum reads $H(k) = \mu_0 + \mu_1 e^{ik}$; this violates the second condition in (A.1) and exhibits pure leftward hopping under OBC. For the sake of clarity, in this Appendix we will explicitly refer to the NH Hamiltonian under OBC (PBC) as $H_{\mathrm{obc}}$ ($H_{\mathrm{pbc}}$).

First, we show that under PBC condition (A.1) is equivalent to $|H_{\mathrm{pbc}}^{-1}| \neq |H_{\mathrm{pbc}}^{-1}|^{\mathrm{T}}$ and then we prove that the equivalence extends to OBC, namely $|H_{\mathrm{obc}}^{-1}| \neq |H_{\mathrm{obc}}^{-1}|^{\mathrm{T}}$; we recall that the modulus is here understood to be taken element-wise. The asymmetry of $|H_{\mathrm{obc}}^{-1}|$ then directly transfers to the susceptibility matrix (13), providing the desired link with non-reciprocity as defined in Sec. 4.

**PBC:** We express $H_{\mathrm{pbc}}^{-1}$ in terms of the singular values decomposition and re-express the result in the site basis, obtaining

$$
\begin{aligned}
H_{\mathrm{pbc}}^{-1} &= \sum_k \frac{1}{\sigma(k)} e^{-i\phi(k)} |k\rangle\langle k| \\
&= \sum_{j,\ell} \frac{1}{N} \sum_k \frac{1}{\sigma(k)} e^{-i\phi(k)} e^{ik(j-\ell)} |j\rangle\langle \ell|.
\end{aligned}
\tag{A.2}
$$

By enforcing $|H_{\mathrm{pbc}}^{-1}|$ to be *symmetric*, we find

$$
|(H_{\mathrm{pbc}}^{-1})_{j,\ell}| = \left| \frac{1}{N} \sum_k \frac{1}{\sigma(k)} e^{-i\phi(k)} e^{ik(j-\ell)} \right| \overset{!}{=} |(H_{\mathrm{pbc}}^{-1})_{\ell,j}| = \left| \frac{1}{N} \sum_{\tilde{k}} \frac{1}{\sigma(k_0 - \tilde{k})} e^{i\phi(k_0 - \tilde{k})} e^{i\tilde{k}(j-\ell)} \right|.
\tag{A.3}
$$

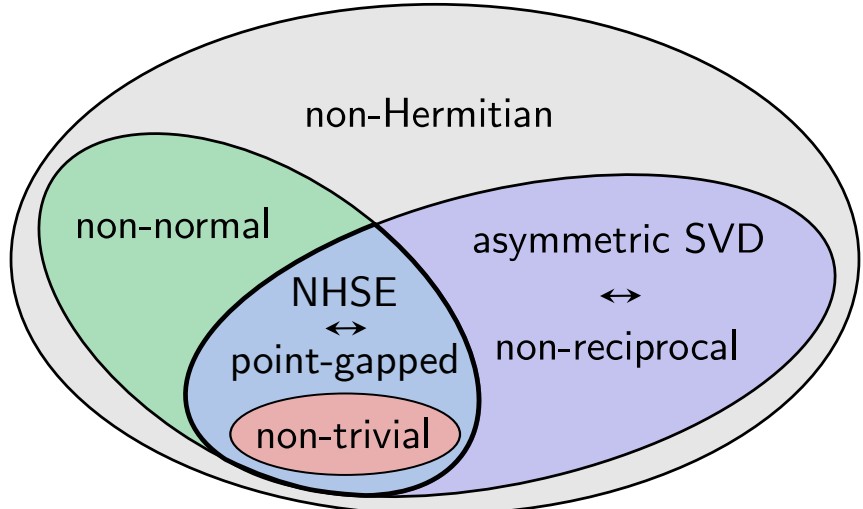

Figure 8: **Relation between non-normality, non-reciprocity and point gap.** The set of non-normal Hamiltonians under OBC and the set of NH Hamiltonians leading to non-reciprocal transport under OBC are two non-empty subsets of the class of NH Hamiltonians. Their intersection is given by the set of point-gapped Hamiltonians, i.e., those NH Hamiltonians featuring a point gap in the PBC spectrum, which coincide with the set of Hamiltonians displaying the NHSE under OBC. Topologically non-trivial Hamiltonians are a strict subset of point-gapped Hamiltonians.

This equality is satisfied if there exists a $k_0$ such that $\sigma(k + k_0) = \sigma(k_0 - k)$ and $\phi(k + k_0) = \phi(k_0 - k)$ for all $k \in [0, 2\pi)$. Equivalently, the converse condition $|H_{\mathrm{obc}}^{-1}| \neq |H_{\mathrm{obc}}^{-1}|^{\mathrm{T}}$ corresponds to either $\sigma(k + k_0) \neq \sigma(k_0 - k)$ or $\phi(k + k_0) \neq \phi(k_0 - k)$ for any $k_0$.

**OBC:** Moving to OBC, we can focus on the topologically trivial case, since we know that in the case of non-trivial topology the localization of the zero singular modes at opposite ends automatically implies non-reciprocity.

In the topologically trivial case, only the bulk contributes to $\chi(\omega)$, see Eq. (40). Under OBC, the bulk singular modes hardly change ($N \gg 1$). At most, plane waves belonging to the same singular value superimpose, with prefactors $\beta(k)$ that satisfy the boundary conditions; note that each singular value appears at least twice, since $H(k)$ forms a closed loop, i.e., $\sigma(k) = \sigma(k')$ for some $k \neq k'$. Therefore, we can write (for $N \gg 1$)

$$H_{\mathrm{obc}}^{-1} \cong \sum_{j,\ell} \frac{1}{N} \sum_k \frac{\beta(k)}{\sigma(k)} e^{\mathrm{i}\phi(k)} e^{\mathrm{i}k(j-\ell)} |j\rangle\langle\ell| \,. \tag{A.4}$$

In order to satisfy the boundary conditions, $\beta(k)$ takes values $\pm 1$ to form anti-symmetric superpositions of singular vectors belonging to the same $\sigma(k) = \sigma(k')$, i.e. $\beta(k) = \pm 1$ and $\beta(k') = \mp 1$. With this condition, the singular vectors have zeros at the edges (this requirement on $\beta(k)$ can be made more rigorous solving the recursion relation under OBC). Since $\beta(k)$ has this simple structure, we can conclude that $|H_{\mathrm{obc}}^{-1}|$ is asymmetric under the same condition (A.1) as $|H_{\mathrm{pbc}}^{-1}|$.

## A.2 The opening of a point gap implies non-normality

Here, we show that the opening of a point gap implies non-normality, namely $[H, H^\dagger] \neq 0$. We recall that an open point gap is defined as a curve in the complex plane that has an interior, see Sec. 6. Here, it is more convenient to use the following equivalent criterion: $H(k)$ has an open

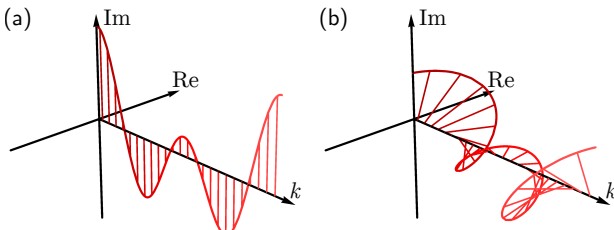

Figure 9: **Condition for the opening of a point gap.** The opening of a point gap can be diagnosed by examining whether the curvature of $H(k)$ is finite for all $k$. The curvature remains finite if $\partial H(k)/\partial k$ does not pass through zero. Zeros as in (a) can be avoided if a phase difference between terms of different $\ell$ in Eq. (A.7) is introduced, as in (b), which ensures that zeros in real and imaginary part do not occur at the same value of $k$.

point gap as long as its curvature remains finite. The intuition behind this criterion is that, since $H(k)$ is a sum of analytic functions and periodic, $H(k) = H(k + 2\pi)$, the only possibility for a diverging curvature is when $H(k)$ has to turn on the spot. This kink only develops in the case of a degenerate spectrum, i.e., when the point gap is not open.

The curvature of $H(k)$ is proportional to $(\partial H(k)/\partial k)^{-1}$, so it diverges when $\partial H(k)/\partial k = 0$. Examining the derivative, we obtain

$$\frac{\partial H(k)}{\partial k} = i \sum_{\ell=1}^{L} \ell (|\mu_\ell| e^{i\phi_\ell} e^{ik\ell} - |\mu_{-\ell}| e^{i\phi_{-\ell}} e^{-ik\ell}). \tag{A.5}$$

We readily see that if the condition

$$|\mu_\ell| \neq |\mu_{-\ell}|, \tag{A.6}$$

is satisfied, the expression is always different from zero, i.e., the point gap is always open. By looking at Eq. (22), the condition we obtained tells us that $H$ is non-normal. If we then examining the case $|\mu_\ell| = |\mu_{-\ell}|$, the expression for the curvature becomes

$$\frac{\partial H(k)}{\partial k} = \sum_{\ell=1}^{L} \ell |\mu_\ell| e^{i(\phi_\ell + \phi_{-\ell})/2} \sin\left(k\ell + \frac{\phi_\ell - \phi_{-\ell}}{2}\right). \tag{A.7}$$

Since the sine term passes through zero for some $k \in [0, 2\pi)$, independent of the values of $\phi_\ell$ and $\phi_{-\ell}$, the only way to prevent $\partial H(k)/\partial k = 0$ is to have a complex prefactor, since this allows the real and imaginary part of $\partial H(k)/\partial k$ to pass through zero at different $k$, see Fig. 9 (a) and (b). In particular, we have to multiply the contributions of at least two different values of $\ell$ by different phase factors $e^{i(\phi_\ell + \phi_{-\ell})/2}$. Hence, for $L \geq 2$, we find as second possibility for the opening of a point gap

$$\phi_\ell + \phi_{-\ell} \neq \phi_{\ell'} + \phi_{-\ell'}, \tag{A.8}$$

which implies non-normality according to Eq. (23) under OBC.

## A.3 The opening of a point gap implies non-reciprocity

Finally, we complete our argument on the basis of the previous proofs. Since an open point gap always implies that the phase $\phi(k) \equiv \text{Arg} H(k)$ is asymmetric with respect some $k_0$, we conclude that the opening of a point gap implies non-reciprocity.

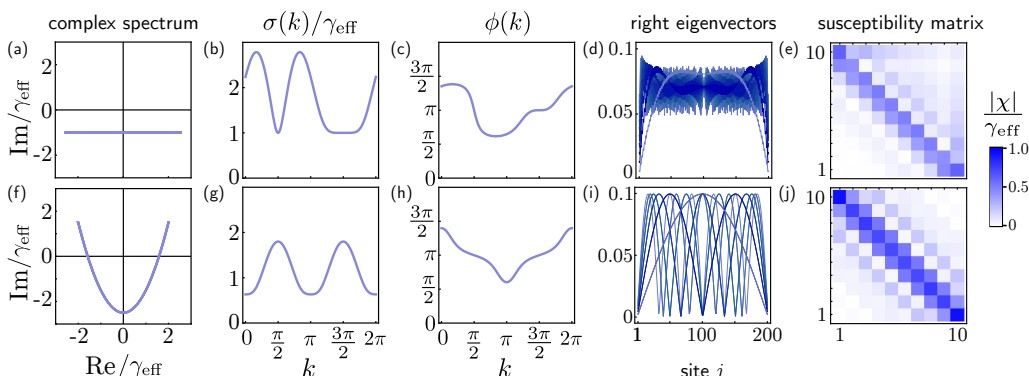

Figure 10: **Counterexamples: non-reciprocity without non-normality and non-normality without non-reciprocity.** (a)-(e) Non-reciprocal but normal system, and (f)-(j) reciprocal but non-normal system. Non-reciprocity, i.e. $|\chi| \neq |\chi|^{\mathrm{T}}$, can be diagnosed from the asymmetry of $\sigma(k)$ and $\phi(k)$ but not from the spectrum which is degenerate in both cases. While the (right) eigenvectors localize weakly in the case of (d) non-reciprocity, they do not localize at all in the reciprocal but non-normal case (i). Only when non-normality and non-reciprocity coincide, see Fig. 8, a point gap opens and the NH skin effect (NHSE) occurs. However, as this comparison shows, non-reciprocity is not always associated with a NHSE. Here, $L = 2$, (a)-(e) $\Lambda_1 = 2$, $\Lambda_2 = 1$, $\mathrm{Arg}\,\Lambda_2 = \pi/2$, $\mathcal{C}_1 = \mathcal{C}_2 = 0$; (f)-(j) $\Lambda_1 = 0.6$, $\mathcal{C}_1 = 0$, $\theta_1 = 0$, $\Lambda_2 = 0$, $\mathcal{C}_2 = 0.8$, $\theta_2 = \pi$.

# B    Examples for normal but non-reciprocal and non-normal but reciprocal systems

Here, we give examples of systems with a closed point gap which are either (i) non-reciprocal but normal, or (ii) non-normal but reciprocal.

We start from (i), illustrating a normal but non-reciprocal system with $L = 2$. Eq. (5) in full generality admits complex coupling constants $J_\ell \neq J_\ell^*$, since gauge freedom only allows us to choose one of the $J_\ell$ real. In our example, we choose $\mathcal{C}_1 = \mathcal{C}_2 = 0$ and $\Lambda_1 \neq 0$, $\Lambda_2 \neq 0$ with $\mathrm{Arg}\,\Lambda_2 = 0$ and $\mathrm{Arg}\,\Lambda_2 = \frac{\pi}{2}$. The phase difference between the coherent hoppings of different range gives rise to constructive and destructive interference, leading to non-reciprocity that is accompanied by an asymmetry in $\sigma(k)$ and $\phi(k)$, while the complex spectrum remains degenerate, see Figs. 10 (a)-(e). Note that this asymmetry cannot be uncovered by looking at the complex spectrum Fig. 10 (a) and only becomes manifest in Figs. 10 (b)-(c). We note that, unlike the exponential localization at a single edge of the NHSE, the eigenvectors only weakly localize and each vector localizes at both ends.

In the second case, we examine a symmetric, but non-normal matrix with $\mu_{-1} = \mu_{+1}$ and $\mu_{-2} = \mu_{+2}$. We obtain this by setting $\Lambda_2 = \mathcal{C}_1 = 0$ but $\Lambda_1 \neq 0$ and $\mathcal{C}_2 \neq 0$ with $\theta_2 = \pi$. Even though $H$ is non-normal according to Eq. (23), the complex spectrum is degenerate, the singular value spectrum and the phase are symmetric, the eigenvectors do not localize at all and the susceptibility matrix is reciprocal, see Figs. 10 (f)-(j).

# C    Non-Hermitian topology for non-zero detuning

In this Appendix, we address the dependence of both the topology and ZSMs on the frequency of the probe. In the main text we focused on the case of resonant probe, i.e. $\delta = 0$ in Eq. (15),

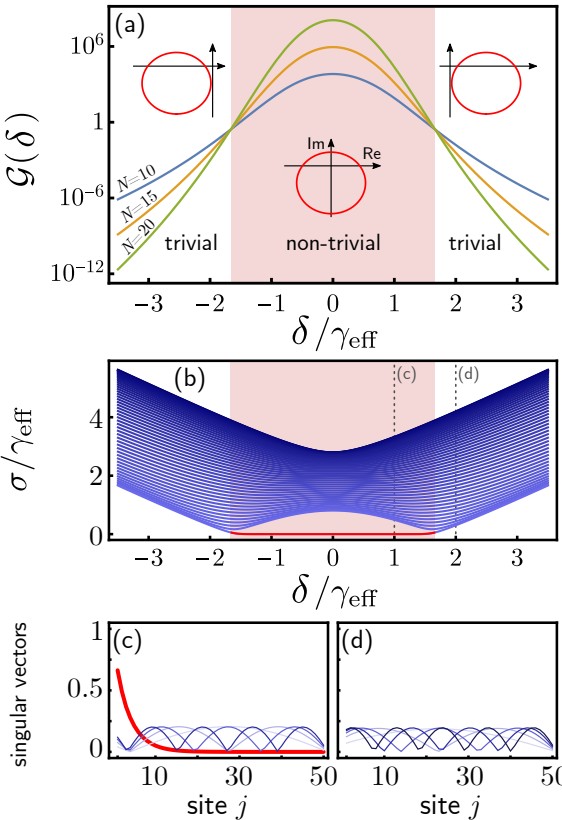

Figure 11: **Frequency dependence of non-Hermitian topology.** (a) For sufficiently small detuning, the winding number remains non-trivial corresponding to gain that grows exponentially with the number of sites; for larger detuning, the winding number becomes trivial and the gain decreases exponentially with system size. (b) Non-trivial topology is associated with a ZSM which appears as we tune the detuning $\Delta$ from strongly off-resonant to resonant. (c) The singular vector corresponding to the ZSM is exponentially localized at the system edge while all other singular vectors remain extended. (d) The trivial phase is characterised by the absence of such ZSMs.

or equivalently $\omega = 0$ in Eq. (13). As explained in the main text, the NH topological invariant (19) depends on the cavity-probe detuning through $\mu_0$, Eq. (15); for clarity, here we consider the case of equal on-site frequencies. Therefore, unlike standard (Hermitian) topology, the notion of NH topology is frequency-dependent, i.e., the frequency at which the system is probed affects its topology. In Fig. 11 we plot the gain and the singular value spectrum Eq. (24) under OBC as a function of cavity-probe detuning $\delta$. From panel (a), we see that the gain grows exponentially with system size in the topologically nontrivial regime, as indicated by the red shaded region, while it is exponentially suppressed in the topologically trivial regime. Indeed, for sufficiently small values of the detuning, we find non-trivial topology under PBC, which coincides with the OBC regime of exponentially growing gain and the presence of one ZSM under OBC, see panel (b); this ZSM is exponentially localized, panel (c). Conversely, for large detuning, we find trivial non-Hermitian topology ($\nu = 0$) under PBC, which corresponds to the absence of ZSMs, i.e., no amplified response; all singular vectors are extended, panel (d). Physically, this means that if a probe signal is too far off resonant, it will not be amplified; a non-amplified response corresponds in turn to a topologically trivial regime. This frequency dependent notion of topology is consistent with our previous work on the topic [28, 90] as well as other works [24, 83, 84].

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
