# Peer review of "Restoration of the non-Hermitian bulk-boundary correspondence via topological amplification"

_SciPost Physics, doi:SciPost Phys. 15, 173 (2023)_

## Round 2 · Referee Report · Anonymous (Referee 1) · 2022-11-23

Strengths

  1. Comprehensive discussion of subtle issues in non-Hermitian physics

  2. Work of great interest for a broad spectrum of physicists from cond-mat to optics and open quantum systems

  3. Reasonably clear presentation in spite of the length and complexity. Of course, some shortening of the text (with no loss of content nor of clarity) would be appreciated.

Weaknesses

  1. A crucial technical issue must be clarified

  2. Referencing to previous literature and putting the results in a wider context may be improved

Report

The manuscript by Brunelli et al. provides an interesting and convincing novel perspective on the role of topology in non-Hermitian systems, mostly of optical nature. The subject is very timely and the manuscript shines new light on a hot topic of present-day research. The results are -in my view- original and, if proven correct, may offer a deep insight on the underlying physics: the study critically compares different approaches to the bulk-boundary correspondance in non-Hermitian systems and provides an unified view on the subject, potentially of great utility for follow-up theory as well as for experiments. I expect it will have a significant impact on a wide community of researchers interested in non-Hermitian models from different points of view, from optics to hydrodynamics, as well as on mathematical physicists. Given the complexity of the work, the authors have done a good job in providing a clear and understandable presentation of their results. Based on these arguments, I find that SciPost will eventually be an ideal venue for its publication once the authors have taken into due consideration the following remarks.

0/ First and most important. As the authors repeatedly mention, the base point for determining the topology plays a crucial role in their theory, as it determines the distinction between "point gap" and "non-trivial topology". My problem is the following. On the one hand, a global shift of the frequencies of the site energies (i.e. of omega_c) can be fully compensated by an identical shift of the probe laser frequency (i.e. the oscillation frequency of \Omega_m(t) or, equivalently, of a_{in,m}(t)), giving at the end identical physical results, which is fine. On the other hand, a shift of the relative frequency of the probe viz. the site energies can change the topology. While the choice of setting the pump to the bare site frequency may appear natural in the Hatano-Nelson model, it is far less obvious in more complex lattices with different on-site frequencies. These arguments, together with some remarks by the authors above eq.(19), push me to think that the topology notion they are introducing is a frequency-dependent one. But if I accept this frequency-dependence of the topology, then I do not understand what is the physical meaning of the ZSMs, since their number can depend on the probe frequency (in contrast to Hermitian lattices, where the BBC is an intrinsic fact of the system and the probe only serves as a measurement tool). And then I would no longer undestand the meaning of the frequency-dependent response (41) calculated by means of the SVD at a given frequency. In a word, I am lost. The authors must clarify this point in full detail before the manuscript can be considered for publication.

1/ Already several decades ago, quite some literature has addressed the interplay of the conservative and dissipative/gain dynamics in hydrodynamical systems, the so-called absolute vs. convective instabilities. These results were then successfully applied to nonlinear optics. The authors should give proper credit to this early literature, quoting the most significant works and, if possible, adding some good discussion on if and how their results may be of interest also for such hydrodynamic problems. For instance, I expect that the sentences around eq.(43) would become clearer if a connection was made to absolute vs. convective instabilities, and I expect that similar improvements may be in order in other points of the ms. As a few entry points into this literature (I am not expert of it, so the authors should make their own bibliographic search), I may mention: Lovergnaux et al., PRL 102 043901 (2004); Pitaevskii et al., PRL 100 160402 (2008); Santagiustina et al., PRL 79, 3633 (1997); Seclì et al., Phys.Rev.Res. 1, 033148 (2019); R. J. Briggs, Electron-Stream Interaction with Plasmas (MIT Press, Cambridge, MA, 1964); P. Huerre and P. A. Monkewitz, Annu. Rev. Fluid Mech. 22, 473 (1990).

1-bis/ Along similar lines, I suggest the authors to explicitly discuss and clarify as much as possible the difference between "directional amplification" and "lasing". I anticipate that several readers may fall into confusion on such subtle distinctions.

2/ I personally do not like the expression "unconditional implementation of...". If I understand correctly what the authors are doing, eqs.(3) are just the equations of motion for the expectation value of the field amplitudes <a_m> under the Master equation (2) (which, for a linear system like the authors' one, are equivalent to a tight-binding reformulation of Maxwell's equation in a gain/loss material). Furthermore, all this ms. deals with classical dynamics, so no reason of using a quantum language. On the of all these arguments, I recomment the authors to look throughout the whole ms. and replace all instances of this misleading terminology.

3/ The authors should define the expression "point gap" the first times they use it: for instance on the LHS column of pag.1 saying "winds in the complex energy plane AS K IS SCANNED ACROSS THE FBZ" (or similar) and then again at the beginning of pag.4. This would help non-expert readers to quickly get to the point of the ms.

4/ At the beginning of pag.5, the authors say that the non-engineered photon decay \gamma is on each input-output waveguide, but in their calculations they assume \gamma to be present on all sites. These two configurations are contradictory as the presence of input/output wavegudies can only affect the sites to which they are connected. The authors should update several points in the text to avoid such a confusion.

5/ For non experts on SVD, the authors may add some reference on the general mathematics of SVD and explicitly mention that the u_j and the v_j form two basis of orthonormal vectors (but do not satisfy any mutual orthogonality condition).

6/ In the caption of fig.3, the quantity \mathcal{C}_2 is called "cooperativity" but its definition before eq.(14) is more like a normalized dissipative coupling. The authors should clarify the terminology.

7/ After eq.39, the authors classify the topology in terms of a Zak phase. Is this really the optimal way of doing, or isn't it better to use the more standard notion of topology related to systems with chiral symmetry? I understand that the two definitions may reduce to similar expressions involving \phi(k) but I am not sure that there are no gauge dependence issues around.

8/ At the end of Sec.XI I am lost about the role of symmetry in the presence of disorder. At the beginning of the section, I understand that the authors are taking a generic disorder that does not necessarily fulfills the chiral symmetry and are showing that the results on the ZSM hold also on this case. Then, in the final paragraph, they seem to require a chiral symmetry. The authors should clarify what they are doing.

8-bis/ in the same last paragraph of Sec.XI, the authors may add parenthesis around tensor products to clarify the formula of the chiral symmetry.

9/ Among the list of experimental systems where this physics can be potentially realized, the authors should also mention exciton-polaritons in micropillar arrays. For completeness, they may also consider adding references to early works on non-Hermitian photonic systems, e.g. by Longhi.

Requested changes

  1. Provide a convincing solution to the conceptual dilemma at point 0/

  2. Provide fair referencing to earlier literature

  3. Take into account all other points

  • validity: good
  • significance: top
  • originality: top
  • clarity: high
  • formatting: perfect
  • grammar: perfect

Author:  Andreas Nunnenkamp  on 2023-05-31  [id 3699]

(in reply to Report 1 on 2022-11-23)

Please find a PDF attached.

Attachment:

Reply1.pdf

---

## Round 2 · Referee Report · Anonymous (Referee 2) · 2023-2-2

Report

The bulk-boundary correspondence (BBC) is a universal feature in the Hermitian system and is essential in understanding and classifying the topological phases. In this paper, the authors generalized this idea to non-hermitian (NH) systems and investigated BBC in NH topological phases. In particular, they focused on a one dimensional energy band with a point gap and showed that there exists a one-to-one correspondence between the bulk topological invariant and the localized modes on the boundary. This paper is well written and includes interesting results. Therefore I recommend it to be published on SciPost after the authors address the following questions:

1 I find the derivation for the BBC is rigorous. However, I do have a practically question about these localized singular vectors. The NH Hamiltonian effectively appears in the open quantum system coupled with an environment. The authors used the singular value spectrum to characterize the topological features of this effective Hamiltonian. Can the authors elaborate on how to measure these singular value spectrum experimentally?

2 If we consider a hermitian topological phases subject to dissipation, we then can obtain a NH Hamiltonian. Do we expect to see such a BBC in this NH Hamiltonian?

3 The symmetry plays a key role in classifying the topological band structure in the Hermitian system and protects the zero modes on the boundary. Do we observe similar phenomena in the NH band theory?
  • validity: high
  • significance: good
  • originality: good
  • clarity: high
  • formatting: excellent
  • grammar: excellent

Author:  Andreas Nunnenkamp  on 2023-05-31  [id 3700]

(in reply to Report 2 on 2023-02-02)

Please find a PDF attached.

Attachment:

Reply2.pdf

---

## Round 3 · Referee Report · Anonymous (Referee 2) · 2023-6-22

Report

The authors addressed all the questions I raised and revised the draft accordingly. The current version can be published on SciPost.

---

## Round 3 · Referee Report · Anonymous (Referee 1) · 2023-7-6

Report

I am very satisfied of the authors' revisions in response to my first report. They have duly taken into account all my remarks, I am (almost) ready to recommend publication.

Before moving into production, the authors may anyway wish considering the following remarks and suggestions:

1/ If I understand correctly, the bottom-rightmost panel of fig.1 (and fig.6(b)), if plotted on a restricted vertical axis [0,1], should display a diagonal feature similar to the one of the central-right of fig.1, with a marked north-east/south-west asymmetry due to non-reciprocity. For the sake of completeness, the authors may include such images in the manuscript, e.g. as insets, and add some corresponding discussion in the text.

2/ In the 3rd column (from left) of fig.1 as well as in the text at the top of the LHS column of pag.4 and in Fig.3(right), the authors use the concept of k vector and of BZ also for the OBC case. They should explain how they are defining k in the case of OBC where translational symmetry is absent.

3/ At the beginning of Sec.IV, the authors speak of an "open quantum system". To avoid any risk of misunderstanding, I would refrain from using the word "quantum" in this work that only deals with classical systems.

4/ A few lines before the beginning of Sec.III, the authors speak of "unit gain" when dealing with the non-topological \nu=0 case. While I agree that in this case the response is way smaller than the one of the topological case \nu\neq 0, I don't see why it should be exactly "unit". And, in fact, in the plots on the rightmost column of Fig.1 the susceptibility is close but not exactly 1. The authors may wish to amend the text to clarify this point.

5/ In the plots of the rightmost column of fig.1 and of fig.6b, the authors should specify which of the x/y axis correspond to the input/output site.

6/ I am (and most likely many other readers are) curious to know the authors' point of view on the possibility of using of the experimental technique of Wang et al. Science 371, 1240 (2021) to study their winding number. I suspect that this technique to reconstruct the k-space dispersion is intrinsically useless in the authors' case where the non-trivial topology forces the PBC dispersion to be dynamically unstable. The authors may add some comments in the manuscript to clarify this interesting point.

---

## Round 3 · Author Response

Dear Editor,

Thank you for handling our submission and for sending us the referee reports.

Referee 1 writes that our manuscript “provides an interesting and convincing novel perspective on the role of topology in non-Hermitian systems”, that it will have “significant impact on a wide community of researchers”, and that “SciPost will eventually be an ideal venue for its publication”. Referee 2 finds our paper is “well written and includes interesting results” and they can therefore, after their questions have been addressed, “recommend it to be published on SciPost”.

In particular, Referee 1 asks us to clarify the dependence of non-hermitian topology, i.e. the number of zero singular modes and the winding number, on frequency or detuning of the probe which we address in a new Appendix C in the revised manuscript. We are grateful that Referee 1 has brought the literature on absolute vs. convective instabilities to our attention which we included in the revised manuscript.

We thank both Referees for their careful review and we hope that they can now recommend our work for publication in SciPost.

Sincerely,
Matteo Brunelli, Clara C. Wanjura, and Andreas Nunnenkamp

---

## Round 3 · List of Changes

Please see our reply to the referee reports.

---

## Round 4 · Author Response

Dear Editor,

In the revised manuscript we address the remaining questions and suggestions in Anonymous Report 2.

Best regards,
The Authors

---

## Round 4 · List of Changes

(1) We added the following sentence in the caption of Fig. 6(b): “Note that the diagonal entries are of order 1, although not discernible from the plot.”

(2) We added the following sentence at the end of Sec. VII, before subsection A (page 9, right column): “To obtain the OBC spectrum in (c)-(e), we write the NH Hamiltonian Eqs. (10) and (11) as the PBC Hamiltonian minus the matrix boundary terms, and express it in the plane-wave basis |𝑘 > where the PBC Hamiltonian is diagonal. We then diagonalize the Hamiltonian < 𝑘|𝐻𝑂𝐵𝐶|𝑘' > and label the eigenstates with 𝑘. The same approach is used for computing the singular value spectrum in Fig. 1 (third column from the left).”

(3) We amended this sentence: “We start from the description of the underlying open quantum system (in order to model explicitly both engineered and non-engineered dissipative processes) and study the dynamics of the classical amplitudes.”

(4) A few lines before the beginning of Sec.III, we amended the sentence to “near-unit gain”.

(5) We added a comment and cited the paper Wang et al. Science 371, 1240 (2021) at the top of page 13, right column: “NH topological amplification entails that the ZSMs are directly measurable in a simple transmission experiment and the topological winding number Eq. (19) can be extracted by counting the number and direction of amplified edge modes, without having to measure the momentum-resolved complex energy band [86].”

---

## Editorial Decision

published